# From Code to Correctness: Closing the Last Mile of Code Generation with Hierarchical Debugging

## Abstract

While large language models have made significant strides in code generation, the pass rate of the generated code is bottlenecked on subtle errors, often requiring human intervention to pass tests, especially for complex problems. Existing LLM-based debugging systems treat generated programs as monolithic units, failing to address bugs at multiple levels of granularity, from low-level syntax errors to high-level algorithmic flaws. In this paper, we introduce *Multi-Granularity Debugger* (MGDebugger), a hierarchical code debugger by isolating, identifying, and resolving bugs at various levels of granularity. MGDebugger decomposes problematic code into a hierarchical tree structure of subfunctions, with each level representing a particular granularity of error. During debugging, it analyzes each subfunction and iteratively resolves bugs in a bottom-up manner. To effectively test each subfunction, we propose an LLM-simulated Python executor, which traces code execution and tracks important variable states to pinpoint errors accurately. Extensive experiments demonstrate that MGDebugger outperforms existing debugging systems, achieving an 18.9% improvement in accuracy over seed generations in HumanEval and a 97.6% repair success rate in HumanEval-Fix. Furthermore, MGDebugger effectively fixes bugs across different categories and difficulty levels, demonstrating its robustness and effectiveness.[1]

## 1 Introduction

Large language models (LLMs) such as GPT-4 (OpenAI, 2023), LLaMA (Touvron et al., 2023), and DeepSeek-Coder (Zhu et al., 2024) have made significant advances in AI-assisted coding tasks (Chen et al., 2021; Lu et al., 2021; Li et al., 2022). Trained on vast corpora of text and code, LLMs can understand and generate code snippets for various programming tasks, ranging from simple data structures to complex algorithmic problems (Li et al., 2022). These models have demonstrated proficiency in tasks such as code completion, bug detection, and even tackling competitive programming challenges.

While the code generated by large models generally meets the requirements, it often contains critical errors that require human intervention to pass tests (Liu et al., 2023b; Dou et al., 2024). This has gradually led to a new development paradigm: large models generate the code, while humans fix it. Therefore, the "last mile", as well as the most crucial step, of code generation is how to efficiently repair the code generated by large models.

Numerous efforts have been made to debug LLM-generated code. The most popular way is to reuse the LLM generator to debug the generated code with the feedback from test case execution (Chen et al., 2023b; Zhong et al., 2024; Hu et al., 2024). While these methods increase the pass rates, they treat the erroneous program as a holistic set of statements (Chen et al., 2023b; Shinn et al., 2023; Zhong et al., 2024; Ding et al., 2024) regardless of the varying types and levels of failures. Failures of test cases arise from different levels of factors, from low-level syntactic errors to high-level algorithmic flaws. A holistic treatment overlooks the internal structure of the code and limits the effectiveness of the debugging systems, especially when dealing with complex programs that need debugging across different modules (Zeller, 2009; Tian et al., 2024).

---

[1]Code and data available at `https://anonymous.4open.science/r/MGDebugger-B388`

In this paper, we introduce Multi-granularity Debugger (MGDebugger), a novel debugging method for LLM-generated code. Instead of treating entire functions as single units, MGDebugger employs a hierarchical, bottom-up strategy to systematically debug code. It begins by decomposing the code into a tree structure of sub-functions, allowing for the isolation of semantic units for independent debugging. Each sub-function is debugged progressively, starting with the most granular ones and working upward to higher-level compositions until the entire code is repaired. To effectively test and debug each subfunction, MGDebugger generates test cases derived from the public test cases of the main function. Then, it employs an LLM-based execution simulator to track changes in key variables, facilitating precise and flexible error identification based on the failed test cases. Through debugging at multiple levels of granularity from the bottom up in a recursive manner, MGDebugger can uncover and rectify bugs that traditional holistic debugging methods might overlook.

Extensive experiments with three models across three benchmarks demonstrate that MGDebugger significantly outperforms existing debugging methods, elevating accuracy from 75.6% to 94.5% on HumanEval (Chen et al., 2021) and achieving a remarkable 97.6% repair success rate on HumanEvalFix (Muennighoff et al., 2023). Ablation studies confirm the vital role of the hierarchical debugging strategy. We also evaluate MGDebugger's effectiveness in handling diverse bug types and varying code lengths, highlighting its robustness and adaptability in real-world coding scenarios. Overall, these results underscore MGDebugger's potential for enhancing the reliability of LLM-generated code.

## 2 RELATED WORK

**Code Generation with LLMs** Recent models such as GPT4 (OpenAI, 2023), Codestral (Mistral AI team, 2024), and DeepSeek-Coder (Zhu et al., 2024) have advanced code generation through instruction tuning and RLHF with mixed code and natural language data (Ziegler et al., 2020; Husain et al., 2020; Rafailov et al., 2023). Code generation with LLMs has been enhanced by various techniques. Some approaches focus on improving the quality of generated code using planning algorithms, transitioning from outlines to detailed implementations (Zhang et al., 2022; Yao et al., 2023; Zelikman et al., 2023; Zhou et al., 2023; Zheng et al., 2023). Other methods sample multiple programs from the same LLM and rank them to identify the best one (Chen et al., 2023a; 2022; Ni et al., 2023). Additionally, some works leverage multi-agent collaboration frameworks to enhance code generation quality (Zhang et al., 2024; Huang et al., 2023a; Dong et al., 2024). These approaches aim to optimize the production of correct code from the outset. By contrast, MGDebugger targets the post-generation phase, focusing on debugging and fixing errors that inevitably arise during the code generation process.

**Repairing LLM-Generated Code** Program repair is a critical aspect of software development, aiming to automatically identify and fix bugs in code (Just et al., 2014; Gupta et al., 2020; Yasunaga & Liang, 2021). There are two main streams of research in repairing code generated by LLMs: (1) training models to repair code (Huang et al., 2023b; Jiang et al., 2024; Ding et al., 2024; Zheng et al., 2024; Moon et al., 2024; Kumar et al., 2024) and (2) providing external feedback to the raw pretrained models to fix code (Jiang et al., 2023; Chen et al., 2023b; Olausson et al., 2023; Zhong et al., 2024; Hu et al., 2024). By contrast to previous work that trains separate models for code repair (Ding et al., 2024; Zheng et al., 2024; Moon et al., 2024), MGDebugger does not require task-specific retraining but takes advantage of the inherent capabilities of pretrained LLMs. This flexibility allows MGDebugger to operate in zero-shot settings, offering a lightweight and scalable alternative. And exploring the ability of LLMs to fix their own code is a promising direction for self-improvement training of the LLMs (Wang et al., 2023; Burns et al., 2023).

MGDebugger falls under the category of work that leverages pretrained models to fix code by reasoning with external feedback. Several recent methods (Zhang et al., 2023; Olausson et al., 2023; Bouzenia et al., 2024; Lee et al., 2024; Xia & Zhang, 2023) utilize execution results from test cases to guide LLMs in code correction. More recent works have explored advanced debugging techniques utilizing LLM's reasoning ability. Reflexion (Shinn et al., 2023) prompts LLMs to reflect on the generated code and uses a memory buffer for iterative refinement. Self-Debugging (Chen et al., 2023b) prompts LLMs to explain or dry run generated programs, known as rubber duck debugging. LDB (Zhong et al., 2024) segments programs into basic blocks, tracking variable values during runtime after each block to verify the correctness against the task description. Although these methods

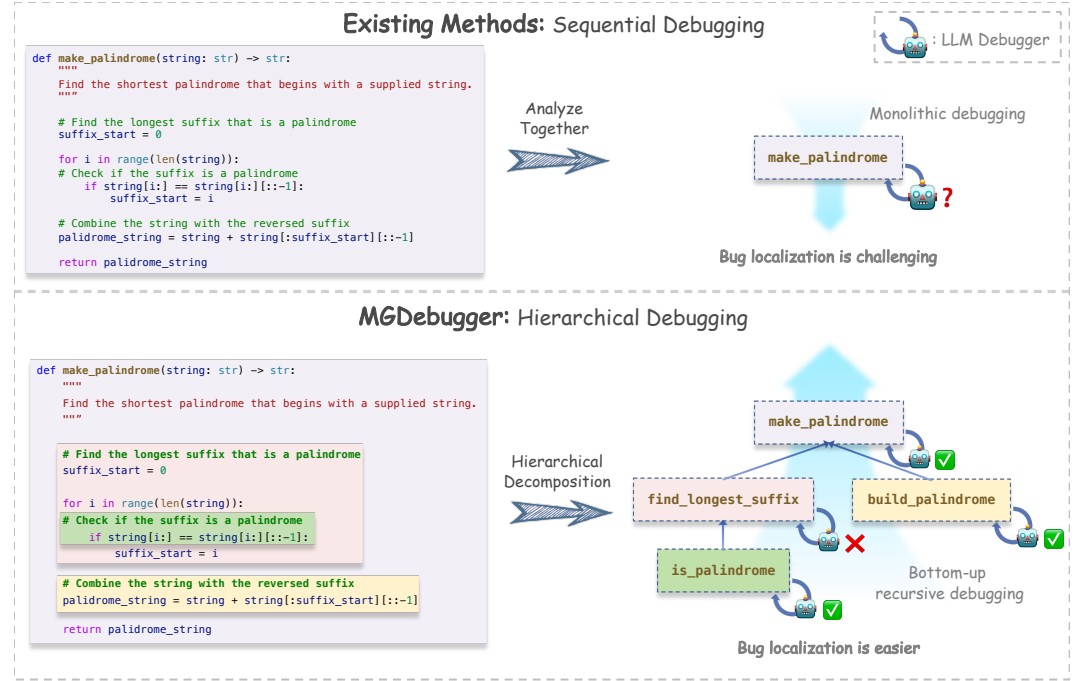

Figure 1: Workflow of MGDebugger compared to existing methods. Existing methods debug the function holistically, making it difficult to pinpoint the bugs. To address this issue, MGDebugger decomposes the code into a hierarchical structure, isolating subfunctions for independent bottom-up debugging. In this way, MGDebugger can identify and fix bugs at multiple levels of granularity, from bottom-level syntax errors to high-level algorithmic flaws. For simplicity, we omit the exact code after decomposition here, and provide the full example in Appendix A.

incorporate detailed execution feedback and iterative refinement, they treat the whole function as a single unit and perform sequential debugging, limiting their effectiveness with complex code (Xia et al., 2023; Hossain et al., 2024). MGDebugger addresses this issue by introducing a hierarchical approach, debugging from low-level errors to high-level flaws. This method ensures a more systematic and accurate debugging process, especially for complex and multifunctional systems.

## 3 METHODOLOGY

### 3.1 OVERVIEW

We present MGDebugger, a novel bottom-up hierarchical debugging method for repairing LLM-generated code. The overall workflow of MGDebugger is illustrated in Figure 1, while the detailed debugging process for each subfunction is depicted in Figure 2.

As shown in Figure 1, MGDebugger begins with *Hierarchical Code Decomposition (Section 3.2)*, which decomposes the input buggy code into a hierarchical structure of subfunctions. This enables systematic identification and resolution of bugs at various levels of granularity. For each subfunction, MGDebugger *Generates Test Case Generation for Subfunctions (Section 3.3)*, deriving private test cases from public test cases of the main function, as illustrated in Figure 2. MGDebugger then executes these test cases and *Debugs Subfunction with LLM-Simulated Execution (Section 3.4)*. The LLM simulates step-by-step code execution for failed test cases, monitoring critical variables and state changes to pinpoint the cause of errors. Once a subfunction has been fixed, MGDebugger updates it in the hierarchical structure and propagates the changes to dependent functions through *Bottom-up Debugging (Section 3.5)*. This hierarchical debugging approach not only tackles different types of bugs at various levels of abstraction but also guarantees a cohesive and systematic debugging process throughout the entire code structure.

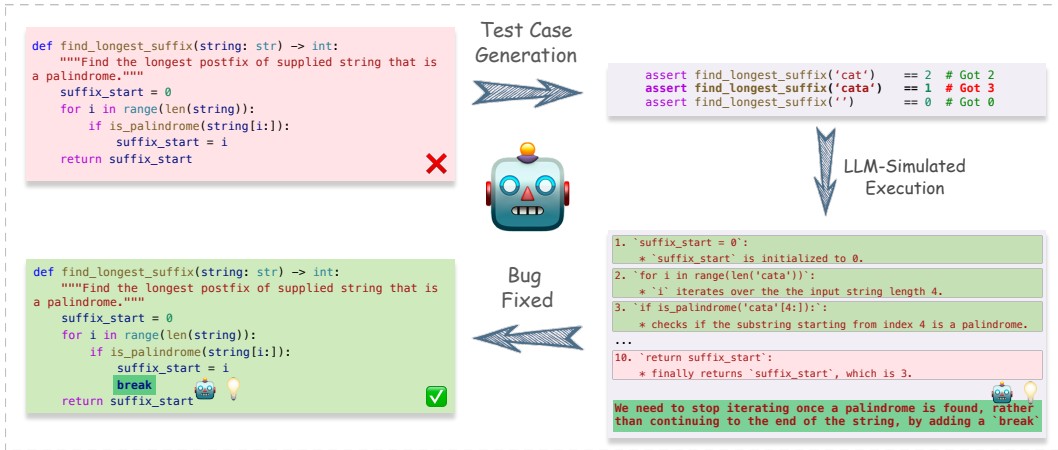

Figure 2: Illustration of the subfunction debugging process. Initially, the LLM generates test cases for the subfunction and collects the results. Subsequently, it simulates the code execution step-by-step, focusing on the change of key variables. This helps the LLM to pinpoint errors accurately and produce a corrected version of the subfunction.

## 3.2 HIERARCHICAL CODE DECOMPOSITION

Modularizing and decomposing complex code into smaller helper subfunctions has been proven to be helpful especially for large functions that are difficult to understand (Jain et al., 2023; Zelikman et al., 2023). To enable hierarchical debugging, we need to transform the input code into a tree-like structure of subfunctions.

Specifically, given an LLM-generated function $f$, we decompose it into a hierarchical structure of subfunctions denoted as $(f_1, ..., f_n)$. These subfunctions can be organized as a tree $f_{\text{root}} = \text{TREE}(f_{\text{root}}, \text{CHILD}(f_{\text{root}}))$, where $f_{\text{root}}$ represents the main function and $\text{CHILD}(f)$ denotes the set of subfunctions directly called by $f$. We leverage an LLM for the decomposition, adhering to three principles: (1) each subfunction represents the minimal reusable unit of code with a specific purpose, (2) higher-level functions call lower-level functions to achieve complex functionality, and (3) the overall structure facilitates isolated testing and debugging. As illustrated in Figure 1, the resulting tree-like structure allows us to isolate logical units of the code, enabling more focused debugging efforts across different levels of granularity (Woodfield et al., 1981; Isazadeh et al., 2017). The prompt template used for code decomposition is provided in Appendix G.1.

## 3.3 GENERATING TEST CASES FOR SUBFUNCTIONS

Having obtained the hierarchy of subfunctions, we aim to verify the correctness of each subfunction. For this purpose, we generate test cases for each subfunction leveraging automatic unit test generation techniques (Wang et al., 2021; Schäfer et al., 2024; Liu et al., 2023a). For each subfunction $f_i \in f_{\text{root}}$, we generate a set of test cases $\mathcal{T}_i$. Following the problem settings from Chen et al. (2023b) and Zhong et al. (2024), we assume that the public test cases for the main function $\mathcal{T}_{\text{pub}}$ have been provided, which is common in most code generation benchmarks (Chen et al., 2021; Hendrycks et al., 2021; Muennighoff et al., 2023)[2]. We can leverage these test cases to derive a set of corresponding test cases for each subfunction.

We employ the same LLM for the test case generation. For each $f_i \in f_{\text{root}}$. The LLM is now prompted to perform the following steps: (1) analyze how the subfunction is used within the main function and how it contributes to the expected outputs in the public test cases; (2) for each public test case, reason through the overall code structure step by step to figure out the input and expected output for the subfunction. This approach ensures that the generated test cases are not only reflective of the subfunction's intended functionality but also contextualized within the constraints provided by the public test cases, enhancing the robustness and relevance of the test cases. The template for generating test cases is provided in Appendix G.2.

---

[2]Otherwise, we can use LLM-generated test cases instead.

---

**Algorithm 1** MGDebugger: Bottom-up Recursive Debugging

**Input:** $f$: Input LLM-generated function; $\mathcal{T}_{\text{pub}}$: Public test cases.
**Output:** $f'$: Debugged $f$.

1: **function** MGDEBUGGER($f, \mathcal{T}_{\text{pub}}$)
2:      **if** $f$ has subfunctions $\{f_1, \ldots, f_n\}$ **then**
3:          **for** $f_i \in f$ **do**             $\triangleright$ Depth-first traversal
4:             $f_i' \leftarrow$ MGDEBUGGER($f_i, \mathcal{T}_{\text{pub}}$)        $\triangleright$ Recursive debugging
5:             $f_i = f_i'$          $\triangleright$ Replace $f_i$ with the debugged version
6:          **end for**
7:      **end if**
8:      $\mathcal{T}_f \leftarrow$ GENTEST($f, \mathcal{T}_{\text{pub}}$)        $\triangleright$ Generate test cases for $f$
9:      $\mathcal{R}_f \leftarrow$ EXEC($f, \mathcal{T}_f$)          $\triangleright$ Execute test cases for $f$
10:     **if** pass $(\mathcal{R}_f, \mathcal{T}_f)$ **then**
11:         **return** $f$        $\triangleright$ Correct function; keep as is
12:     **else**
13:         $f' \leftarrow$ DEBUG($f, \mathcal{T}_f, \mathcal{R}_f$)     $\triangleright$ Debug function $f$ based on test results $\mathcal{R}_f$
14:         **return** $f'$        $\triangleright$ Return the corrected code
15:     **end if**
16: **end function**

---

### 3.4 DEBUGGING SUBFUNCTIONS WITH LLM-SIMULATED EXECUTION

With the generated test cases, we debug each subfunction by running them on the test case inputs, obtaining the results, and comparing these results against the expected outcomes in the test cases. When a failed test case is identified, we fix the corresponding subfunction and produce a corrected version.

One straightforward way to implement this process is to use an external Python executor to monitor runtime variable values (Zhong et al., 2024). However, when debugging high-level functions, tracking variable values within lower-level subfunctions is often unnecessary, as their correctness is ensured by the bottom-up debugging methodology. Furthermore, directly collecting all execution traces from the external debugger can add unnecessary overhead and complexity to the process.

Inspired by the methodology in Li et al. (2023), we propose an LLM-simulated code executor, which prompts LLM to act as a Python interpreter and track the code execution. As shown in Figure 2, we request the LLM to simulate the execution process, reasoning about key variables and their states at each step, and thoroughly analyzing the failed test cases. This eliminates the need for an external debugger, offering a more flexible and efficient debugging solution. In addition, the LLM can accurately identify where errors occur and grasp their surrounding context. The LLM prompt for the debugging process is detailed in Appendix G.3.

### 3.5 BOTTOM-UP DEBUGGING

Having introduced code decomposition and the debugging process for each subfunction, we now outline the overall debugging workflow.

We initiate the process by calling MGDebugger on the main function with the decomposed code $f_{\text{root}}$ and the set of public test cases $\mathcal{T}_{\text{pub}}$. MGDebugger traverses the hierarchical structure in a depth-first manner, recursively debugging each subfunction before moving on to the higher-level functions. For each specific subfunction, MGDebugger generates relevant test cases and debugs the function based on the results. When a fix is identified, MGDebugger updates the function and propagates the changes to the dependent functions. This recursive, bottom-up strategy systematically addresses bugs , beginning with the most granular levels and progressively advancing through the function hierarchy. This method accommodates various types of bugs at different abstraction levels, from low-level syntax errors to high-level logical flaws, by focusing on one level of the hierarchy at a time and building up the corrected code in a structured manner. The detailed algorithm is presented in Algorithm 1.

# 4 EXPERIMENTS

## 4.1 SETUP

**Models** We select three state-of-the-art LLMs ranging from 7B to 22B parameters as backbones for code generation and debugging: CodeQwen1.5 (7B) (Bai et al., 2023), DeepSeek-Coder-V2-Lite (16B) (Zhu et al., 2024), and Codestral (22B) (Mistral AI team, 2024). Please refer to Appendix C for our implementation details.

**Datasets** We conduct experiments on three datasets. HumanEval (Chen et al., 2021) and MBPP (Austin et al., 2021) are two widely used benchmarks for evaluating code generation systems with 164 and 500 problems, respectively. The HumanEvalFix dataset (Muennighoff et al., 2023) consists of 164 buggy functions with six different bug categories: value misuse, missing logic, excess logic, operator misuse, variable misuse, and function misuse. The detailed explanations and distribution of bug categories can be found in Appendix B.

**Metrics** We adopt two metrics to evaluate our method: 1) *Accuracy* (Chen et al., 2023b; Zhong et al., 2024), which measures the overall proportion of correct code samples among all generated code samples after debugging. A code is correct iff it passes all private test cases assigned to it. 2) *Repair Success Rate* (RSR) (Yasunaga & Liang, 2021), which refers to the proportion of fixed code samples to the total number of buggy code samples.

**Baselines** We compare MGDebugger with eight state-of-the-art methods for debugging LLM-generated code. 1) *Simple Feedback* is a basic baseline that informs the LLM that the code is incorrect and asks it to fix the issue. 2) *Self-Edit* (Zhang et al., 2023) prompts the LLM to edit the code based on the execution results of the test cases. 3) *Self-Debugging* (Chen et al., 2023b) has two variants: *Self-Debugging (Expl.)* prompts the LLM to explain the generated code line-by-line, while *Self-Debugging (Trace)* asks the LLM to dry run the code for debugging. 4) *LDB* (Zhong et al., 2024) segments the code into basic *blocks*, *functions* or *lines*, and tracks variable values during runtime after each block to verify correctness against the task description. 5) *Reflexion* (Shinn et al., 2023) asks the LLM to reflect on the previous code given execution results and uses a memory buffer to enable iterative refinement.

## 4.2 MAIN RESULTS

The results in Table 1 show that MGDebugger consistently outperforms existing approaches across all models and datasets. Specifically, MGDebugger achieves the highest accuracy improvements, with gains of +15.3% to +18.9% on HumanEval and +11.4% to +13.4% on MBPP. These improvements are particularly notable when compared to baseline methods such as Self-Debugging (Expl.) and Reflexion, which also incorporate external feedback but exhibit lower gains in accuracy and RSR. The strong results across models of varying sizes highlight the adaptability of MGDebugger to different LLM architectures.

Moreover, MGDebugger demonstrates remarkable debugging capabilities, particularly with DeepSeek-Coder-V2-Lite (16B) and Codestral (22B), where it achieves an accuracy of 94.5% on the HumanEval dataset, the highest score among all methods. This is especially impressive considering that MGDebugger operates in a zero-shot setting without task-specific retraining. This result illustrates the inherent debugging ability of larger LLMs with MGDebugger. Additionally, the method's performance on MBPP, achieving an RSR of up to 41.1% with smaller models like Code-Qwen1.5 (7B), further underscores its robustness. In general, these results validate MGDebugger as a highly effective and scalable debugging method for LLM-generated code.

## 4.3 ABLATION STUDY

To understand the contribution of each component in MGDebugger and validate our design choices, we conduct an ablation study by systematically removing key components of our method: hierarchical code decomposition, LLM-simulated execution, and test case generation for subfunction debugging. Each variant is evaluated on both the HumanEval and MBPP datasets using the DeepSeek-Coder-V2-Lite model.

Table 1: Results of MGDebugger and other methods on HumanEval and MBPP. Acc.: Accuracy, $\Delta$: Improvement over baseline (No-Debugging), RSR: Repair Success Rate.

| Model | Method | Dataset | | | | | |
| | | HumanEval | | | MBPP | | |
| | | Acc. (%) | $\Delta$ Acc. (%) | RSR (%) | Acc. (%) | $\Delta$ Acc. (%) | RSR (%) |
|---|---|---|---|---|---|---|---|
| DeepSeek-Coder-V2-Lite | No-Debugging | 76.8 | – | – | 67.2 | – | – |
| | Simple Feedback | 82.3 | +5.5 | 23.7 | 69.4 | +2.2 | 6.7 |
| | Self-Edit | 82.9 | +6.1 | 26.3 | 71.2 | +4.0 | 12.2 |
| | LDB (Block) | 84.1 | +7.3 | 31.6 | 74.0 | +6.8 | 20.7 |
| | LDB (Line) | 82.3 | +5.5 | 23.7 | 71.8 | +4.6 | 14.0 |
| | LDB (Function) | 81.7 | +4.9 | 21.1 | 72.6 | +5.3 | 16.5 |
| | Self-Debugging (Expl.) | 87.2 | +10.4 | 44.7 | 73.4 | +6.2 | 18.9 |
| | Self-Debugging (Trace) | 86.0 | +9.2 | 39.5 | 72.6 | +5.3 | 16.5 |
| | Reflexion | 90.9 | +14.1 | 60.5 | 76.6 | +9.4 | 28.7 |
| | **MGDebugger** | **94.5** | **+17.7** | **76.3** | **80.0** | **+12.8** | **39.0** |
| CodeQwen1.5 | No-Debugging | 76.2 | – | – | 67.4 | – | – |
| | Simple Feedback | 85.4 | +9.2 | 38.5 | 74.0 | +6.6 | 20.2 |
| | Self-Edit | 84.1 | +7.9 | 33.3 | 75.0 | +7.6 | 23.3 |
| | LDB (Block) | 79.3 | +3.1 | 12.8 | 72.8 | +5.4 | 16.6 |
| | LDB (Line) | 79.9 | +3.7 | 15.4 | 72.6 | +5.2 | 16.0 |
| | LDB (Function) | 80.5 | +4.3 | 17.9 | 72.8 | +5.4 | 16.6 |
| | Self-Debugging (Expl.) | 87.8 | +11.6 | 48.7 | 77.4 | +10.0 | 30.7 |
| | Self-Debugging (Trace) | 84.8 | +8.6 | 35.9 | 76.8 | +9.4 | 28.8 |
| | Reflexion | 87.8 | +11.6 | 48.7 | 78.6 | +11.2 | 34.4 |
| | **MGDebugger** | **91.5** | **+15.3** | **64.1** | **80.8** | **+13.4** | **41.1** |
| Codestral | No-Debugging | 75.6 | – | – | 65.4 | – | – |
| | Simple Feedback | 88.4 | +12.8 | 52.5 | 71.6 | +6.2 | 17.9 |
| | Self-Edit | 86.0 | +10.4 | 42.5 | 75.8 | +10.4 | 30.0 |
| | LDB (Block) | 83.5 | +7.9 | 32.5 | 72.2 | +6.8 | 19.7 |
| | LDB (Line) | 83.5 | +7.9 | 32.5 | 71.8 | +6.4 | 18.5 |
| | LDB (Function) | 82.3 | +6.7 | 27.5 | 72.0 | +6.6 | 19.1 |
| | Self-Debugging (Expl.) | 89.6 | +14.0 | 57.5 | 76.4 | +11.0 | 31.8 |
| | Self-Debugging (trace) | 84.1 | +8.5 | 35.0 | 73.6 | +8.2 | 23.7 |
| | Reflexion | 86.6 | +11.0 | 45.0 | 75.2 | +9.8 | 28.3 |
| | **MGDebugger** | **94.5** | **+18.9** | **77.5** | **76.8** | **+11.4** | **32.9** |

Table 2: Ablation study results for DeepSeek-Coder-V2-Lite. Acc.: Accuracy, $\Delta$ Acc.: Improvement over baseline (No-Debugging), RSR: Repair Success Rate.

| Method | HumanEval | | | MBPP | | |
| | Acc. (%) | $\Delta$ Acc. (%) | RSR (%) | Acc. (%) | $\Delta$ Acc. (%) | RSR (%) |
|---|---|---|---|---|---|---|
| **MGDebugger** | **94.5** | **+17.7** | **76.3** | **80.0** | **+12.8** | **39.0** |
| - w/o Hierarchical Debugging | 89.0 | +12.2 | 52.6 | 78.2 | +11.0 | 33.5 |
| - w/o Simulated Execution | 90.2 | +13.4 | 61.3 | 79.2 | +12.0 | 36.6 |
| - w/o Test Case Generation | 90.9 | +14.1 | 60.5 | 79.2 | +12.0 | 36.6 |
| No-Debugging | 76.8 | – | – | 67.2 | – | – |

As shown in Table 2, each component of MGDebugger plays a crucial role in the overall effectiveness of the method. Among them, the hierarchical debugging strategy is the most impactful component. By ablating this strategy, the repair success rate drops significantly from 76.3% to 52.6% on HumanEval and from 39.0% to 33.5% on MBPP. This result highlights the importance of the hierarchical approach in systematically identifying and fixing bugs at different granularity levels. Additionally, the LLM-simulated execution and test case generation for subfunctions also facilitate debugging the decomposed code, yielding substantial improvements in accuracy and repair success rates. These results underscore the effectiveness of MGDebugger's design choices and the importance of its hierarchical debugging strategy.

Table 3: Performance (RSR) on different bug categories in HumanEvalFix with different models. The best and second-best scores are highlighted in bold and underline, respectively.

| Method | Value | Missing Logic | Excess Logic | Operator | Variable | Function | Overall |
|---|---|---|---|---|---|---|---|
| DeepSeek-Coder-V2-Lite | | | | | | | |
| Simple Feedback | 84.9 | 96.0 | 80.7 | 78.3 | 86.4 | 87.5 | 85.4 |
| Self-Edit | 78.8 | 92.0 | 80.7 | 82.6 | 84.1 | 62.5 | 82.3 |
| LDB (Block) | 69.7 | 96.0 | 74.2 | 87.0 | 86.4 | 62.5 | 81.1 |
| LDB (Line) | 63.6 | 84.0 | 67.7 | 73.9 | 84.1 | 62.5 | 74.4 |
| LDB (Function) | 69.7 | 88.0 | 71.0 | 87.0 | 77.3 | 62.5 | 76.8 |
| Self-Debugging (Expl.) | 66.7 | 80.0 | 64.5 | 78.3 | 86.4 | 50.0 | 74.4 |
| Self-Debugging (Trace) | 81.8 | 88.0 | 71.0 | 78.3 | 79.6 | 75.0 | 79.3 |
| Reflexion | 90.9 | 100.0 | 90.3 | 91.3 | 86.4 | 100.0 | 91.5 |
| **MGDebugger** | 87.9 | 100.0 | 100.0 | 100.0 | 100.0 | 100.0 | **97.6** |
| CodeQwen1.5 | | | | | | | |
| Simple Feedback | **81.8** | 92.0 | 87.1 | 69.6 | 81.8 | 87.5 | 82.9 |
| Self-Edit | 72.7 | 92.0 | 80.7 | 65.2 | **86.4** | 87.5 | 80.5 |
| LDB (Block) | 36.4 | 72.0 | 51.6 | 60.9 | 63.6 | 62.5 | 56.7 |
| LDB (Line) | 36.4 | 76.0 | 45.2 | 56.5 | 54.6 | 50.0 | 52.4 |
| LDB (Function) | 27.3 | 60.0 | 51.6 | 56.5 | 59.1 | 62.5 | 51.2 |
| Self-Debugging (Expl.) | 69.7 | 92.0 | **90.3** | 69.6 | 77.3 | 62.5 | 78.7 |
| Self-Debugging (Trace) | 72.7 | 72.0 | 80.6 | 69.6 | 70.5 | 75.0 | 73.2 |
| Reflexion | 66.7 | 88.0 | 80.6 | 91.3 | **86.4** | 75.0 | 81.7 |
| **MGDebugger** | 78.8 | **96.0** | 87.1 | **95.7** | 84.1 | **100.0** | **87.8** |
| Codestral | | | | | | | |
| Simple Feedback | 75.8 | 92.0 | 67.7 | 82.6 | 84.1 | 62.5 | 79.3 |
| Self-Edit | 78.8 | **100.0** | 80.7 | 87.0 | 84.1 | **87.5** | 85.4 |
| LDB (Block) | 66.7 | 92.0 | 67.7 | 82.6 | 81.8 | **87.5** | 78.1 |
| LDB (Line) | 63.6 | 92.0 | 64.5 | 82.6 | 81.8 | 75.0 | 76.2 |
| LDB (Function) | 57.6 | 88.0 | 67.7 | **91.3** | 75.0 | 75.0 | 74.4 |
| Self-Debugging (Expl.) | 75.8 | 96.0 | 83.9 | 87.0 | 90.9 | **87.5** | 86.6 |
| Self-Debugging (Trace) | 57.6 | 84.0 | 64.5 | 73.9 | 81.8 | 75.0 | 72.6 |
| Reflexion | 69.7 | 88.0 | 61.3 | 82.6 | 88.6 | 75.0 | 78.0 |
| **MGDebugger** | **87.9** | **100.0** | **87.1** | 82.6 | **95.5** | 75.0 | **90.2** |

## 4.4 DEBUGGING DIFFERENT TYPES OF BUGS

To assess the versatility and effectiveness of MGDebugger across various bug categories, we carry out experiments using the HumanEvalFix dataset, which is specifically designed to evaluate code debugging performance. The dataset involves six distinct bug categories: value misuse, missing logic, excess logic, operator misuse, variable misuse, and function misuse, allowing us to examine how effectively MGDebugger addresses different types of programming errors compared to existing methods. The detailed explanations of each bug category are available in Appendix B.

Table 3 presents the RSRs across various bug categories. We observe that MGDebugger consistently outperforms other methods with significantly higher overall accuracies. And MGDebugger achieves a remarkable repair success rate of 97.6% using DeepSeek-Coder, with 100% success rates in all bug categories except for value misuse. This is particularly notable given the complexity and diversity of the bugs in the dataset. This highlights the effectiveness of the hierarchical debugging strategy.

Looking into details of different bug categories, MGDebugger shows a strong advantage in debugging bottom-level bugs, such as missing logic and excess logic. Missing logic refers to situations where essential code is omitted, preventing the solution from functioning correctly. Excess logic, on the other hand, involves unnecessary code that can lead to mistakes and confusion (Muennighoff et al., 2023). Other methods often struggle to identify and address these underlying issues because they treat the code holistically. This can lead to confusion over bottom-level details when dealing with complex logical errors. By contrast, the hierarchical decomposition in MGDebugger allows it to focus on different levels of code granularity. This enables more effective identification and correction of bugs. These results demonstrate the robustness and versatility of MGDebugger across various bug types.

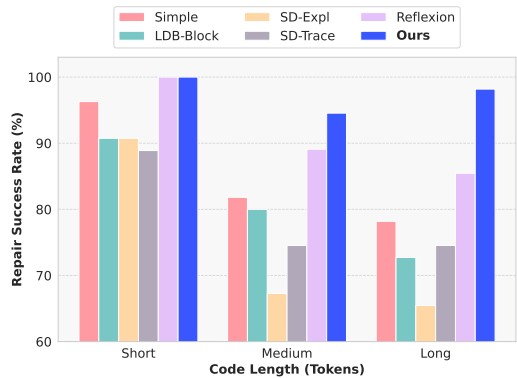
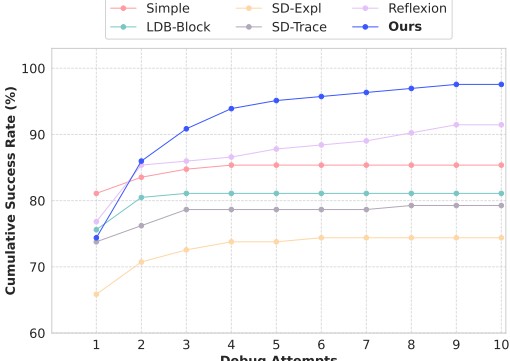

Figure 3: Repair success rate of different methods when debugging code of different lengths on HumanEvalFix with DeepSeek-Coder. MGDebugger consistently outperforms other methods across different code lengths, especially in long codes.

Figure 4: Impact of the number of debugging attempts on the cumulative repair success rate of MGDebugger and other methods on HumanEvalFix with DeepSeek-Coder. MGDebugger continues to improve with more debug attempts and achieves the highest success rate.

### 4.5 DEBUGGING CODE WITH VARYING LENGTH

We further assess the versatility of MGDebugger in debugging code of varing lengths (i.e., number of tokens), since code length often correlates with complexity and debugging challenges. We categorize code snippets from the HumanEvalFix dataset into short, medium, and long groups, ensuring equal sample sizes. We subsequently analyze the RSR scores obtained by MGDebugger and baselines when using DeepSeek-Coder as the backbone LLM.

The results are presented in Figure 3. We can observe that as the code length increases, most methods experience an obvious decrease in performance due to the increased complexity. We note that MGDebugger consistently outperforms other methods in different code lengths and especially excels in debugging longer and more complex code snippets. This showcases the scalability and robustness of MGDebugger in handling code of varying lengths and complexities. The results on other two datasets are available in Appendix D, where MGDebugger also consistently outperforms other methods across different code lengths.

### 4.6 IMPACT OF DEBUG ATTEMPTS

Another important factor for LLM-based debugging is the number of debugging attempts. Iterative debugging allows LLMs to refine their corrections over multiple passes, potentially leading to better outcomes. We aim to assess MGDebugger's ability to improve over successive iterations. Following Zhong et al. (2024), we vary the number of debugging attempts from 1 to 10 using the HumanEvalFix dataset and DeepSeek-Coder.

The results in Figure 4 show that MGDebugger achieves the highest cumulative RSR score among all methods, highlighting its ability to continually refine its debugging over multiple attempts. In particular, while most methods plateau after the first few debug attempts, MGDebugger and Reflexion continue to improve with more iterations. This result underscores the great potential of MGDebugger for iterative and comprehensive debugging, making it a promising solution for complex and challenging code repair tasks. The results on the other two datasets are available in Appendix E, where MGDebugger outperforms other methods from the first attempt and continues to improve with great potential.

### 4.7 CASE STUDY

We perform a qualitative analysis of how MGDebugger effectively identifies and corrects buggy parts compared to baseline methods. Figure 5 shows an example of debugging code snippets from the HumanEvalFix dataset using MGDebugger and other representative methods, with DeepSeek-

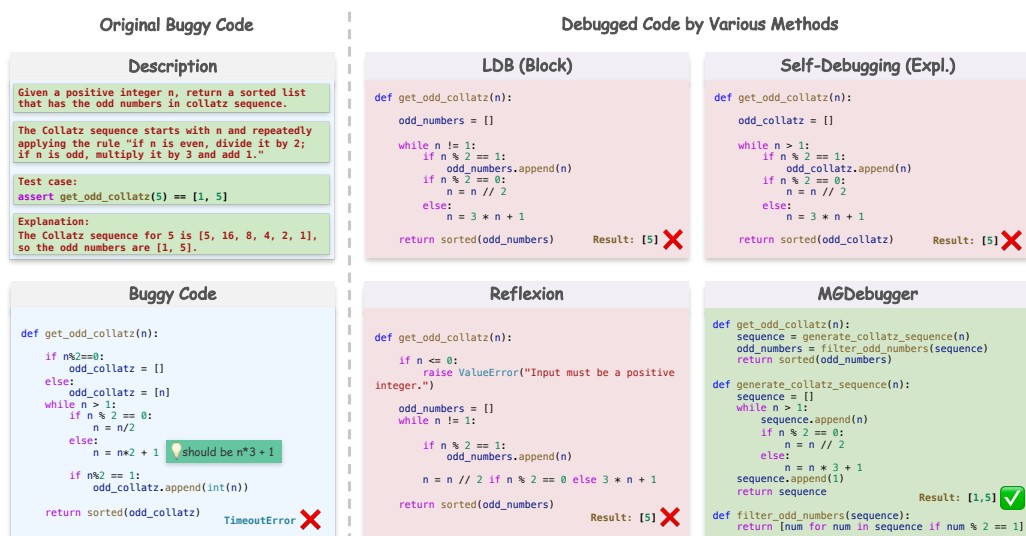

Figure 5: Examples of code debugging by various methods on HumanEvalFix with DeepSeek-Coder. The three baseline methods fix the original bug but introduce new bugs that will miss the last "1" in the results. By contrast, MGDebugger successfully identifies and corrects the bug after decomposing the code into clear subfunctions for separate debugging.

Coder-V2-Lite as the backbone LLM. The original buggy solution computes the Collatz sequence with an incorrect computation logic of $n = n \times 2 + 1$. While other methods correct the computation to $n = n \times 3 + 1$, they introduce a new bug that misses the last "1" in the Collatz sequence. This is possibly because they get distracted by the need to filter odd numbers, and thus move the operation of appending the number to the results before updating $n$. MGDebugger excelled by decomposing the problem into distinct subfunctions: sequence generation and odd number filtering. By debugging each subfunction independently, MGDebugger ensured comprehensive error correction, including the subtle requirement of incorporating 1 into the Collatz sequence. This approach demonstrates MGDebugger's ability to handle complex, multi-step problems more effectively than holistic debugging methods. Additionally, it highlights MGDebugger's ability to not only fix bugs but also restructure code for enhanced clarity and correctness, demonstrating its potential in improving the quality of LLM-generated code. More examples and analysis on the three datasets can be found in Appendix F.

## 5 CONCLUSION

In this paper, we introduced MGDebugger, a novel hierarchical code debugging framework that systematically fixes bugs at multiple levels of granularity. By decomposing complex code into a hierarchical structure, generating targeted test cases and employing LLM-simulated execution, MGDebugger effectively identifies and fixes bugs ranging from syntax errors to logical flaws in a bottom-up manner. Experiments across various models and datasets demonstrate MGDebugger's superior performance over existing methods, particularly in handling complex logical errors and longer code snippets.

Future work can build upon this foundation to develop more advanced code generation and debugging methodologies. One direction is to extend MGDebugger to handle more complex bugs and code structures, such as multi-file projects and codebase with multiple dependencies. Another direction is to explore the collaboration of hierarchical code generation approaches such as Parsel (Zelikman et al., 2023) with hierarchical debugging, enabling end-to-end code generation and debugging systems. Furthermore, integrating MGDebugger into self-training systems to correct outputs from base models, then retraining the base models with the corrected data, could potentially improve their performance iteratively (Gulcehre et al., 2023).

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

## A    DETAILED HIERARCHICAL DECOMPOSITION EXAMPLE

We provide the detailed illustration of the hierarchical decomposition process in MGDebugger, as shown in Figure 6, which has been simplified for illustration in Figure 1. For the original function "make_palindrome", we decompose it into three minimal reusable subfunctions. And the relationships between subfunctions are naturally captured in the hierarchical structure based on the function calls. This hierarchical decomposition allows MGDebugger to systematically analyze and debug the code at different levels of granularity, leading to more effective identification and correction of bugs.

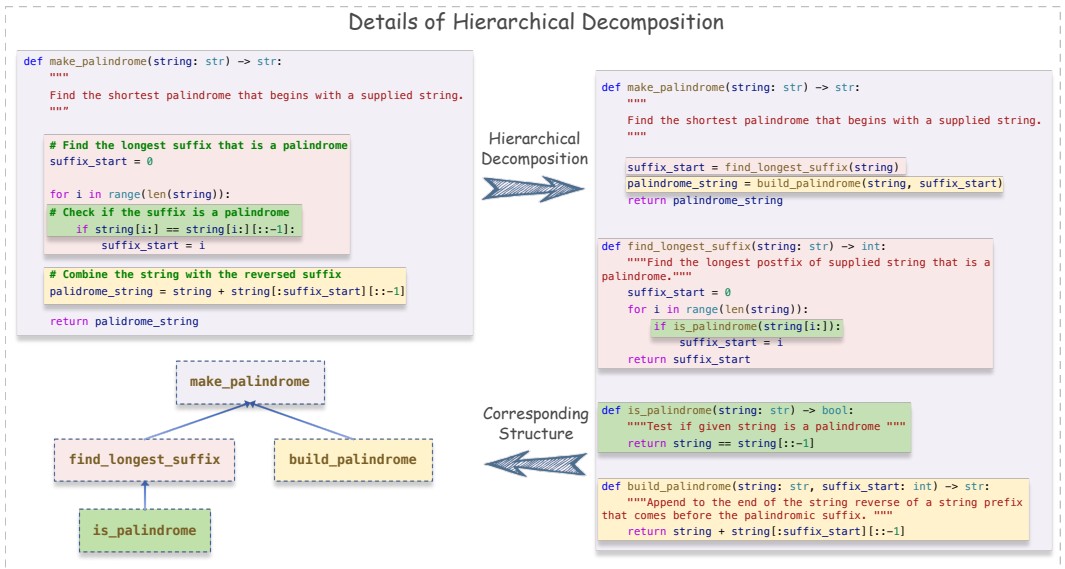

Figure 6: Detailed illustration of the hierarchical decomposition process in MGDebugger. The original code is decomposed into multiple sub-functions, each representing a significant step or logical block. The relationships between sub-functions are naturally captured in the hierarchical structure based on the function calls.

## B    HUMANEVALFIX DATASET

To access the ability of MGDebugger in debugging code with different types of bugs, we use the HumanEvalFix dataset (Muennighoff et al., 2023), which consists of 164 buggy functions across six programming languages, each provided with solutions and unit tests. For our experiments, we focus on the Python subset of the dataset. The buggy functions are categorized into six types of bugs: value misuse, missing logic, excess logic, operator misuse, variable misuse, and function misuse. Table 4 shows the distribution and explanations of these bug types within the HumanEvalFix dataset.

Table 4: Distribution and explanations of bugs in the HumanEvalFix dataset.

| Bug Category | Explanation | Count |
|---|---|---|
| Value Misuse | An incorrect value is used | 44 |
| Missing Logic | Misses code needed to solve the problem | 33 |
| Excess Logic | Contains excess code leading to mistakes | 31 |
| Operator Misuse | An incorrect operator is used | 25 |
| Variable Misuse | An incorrect variable is used | 23 |
| Function Misuse | An incorrect function is used | 8 |

## C  IMPLEMENTATION DETAILS

We generate seed programs for HumanEval and MBPP using the BigCode Evaluation Harness framework[3]. The specific versions of models used in our experiments are DeepSeek-Coder-V2-Lite-Instruct[4], CodeQwen1.5-7B-Chat[5], and Codestral-22B-v0.1[6]. All experiments are conducted on NVIDIA A100 GPUs with 80GB memory. During debugging, we use the vLLM engine[7] to serve the LLMs, setting the maximum token length according to each LLM's max length. Following Zhong et al. (2024), we limit the maximum number of debugging iterations to 10 for all methods. Additionally, the sampling temperature is set to 0.8 in MGDebugger.

To obtain visible test cases for HumanEval and HumanEvalFix, we extract the given visible test cases from the task description. For MBPP, we use the first test case of each problem as the visible test case and use the rest as hidden test cases, in line with the settings referenced from Chen et al. (2023b) and Zhong et al. (2024).

## D  DEBUGGING CODE WITH VARYING LENGTHS ON HUMANEVAL AND MBPP

To further demonstrate the robustness of MGDebugger in handling code of varying lengths, we present examples from MBPP and HumanEval. Similar to the examples provided for HumanEval-Fix, we categorize the problems into short, medium, and long groups based on their code lengths, and we measure the repair success rates of MGDebugger and other baseline methods. All methods are built upon DeepSeek-Coder-V2-Lite. As is observed in Figure 7 and Figure 8, MGDebugger consistently outperforms other methods across different code lengths, especially in longer codes. This result demonstrates the scalability and robustness of MGDebugger in handling code of varying lengths and complexities again.

## E  IMPACT OF DEBUG ATTEMPTS ON HUMANEVAL AND MBPP

We also investigate the impact of debug attempts on the cumulative repair success rate of MGDebugger and other methods on HumanEval and MBPP. As shown in Figure 9 and Figure 10, MGDebugger continues to improve with more debug attempts and achieves the highest success rate among all methods. Different from the results on HumanEvalFix that MGDebugger starts to ourperform other methods after the first attempt, MGDebugger significantly outperforms other methods from the beginning to the end on HumanEval and MBPP. This result highlights the effectiveness of MGDebugger in iterative and comprehensive debugging, making it a promising solution for complex and challenging code repair tasks.

---

[3] `https://github.com/bigcode-project/bigcode-evaluation-harness`
[4] `https://huggingface.co/deepseek-ai/DeepSeek-Coder-V2-Lite-Instruct`
[5] `https://huggingface.co/Qwen/CodeQwen1.5-7B-Chat`
[6] `https://huggingface.co/TechxGenus/Codestral-22B-v0.1-GPTQ`
[7] `https://github.com/vllm-project/vllm`

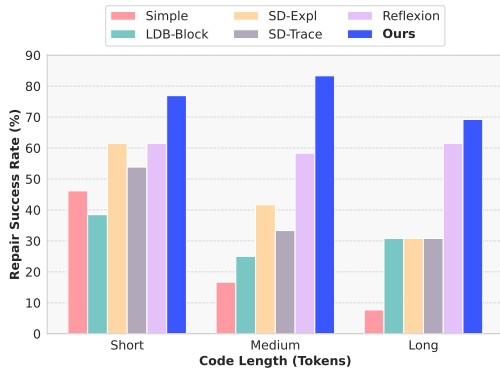

Figure 7: Repair success rate of different methods when debugging code of different lengths on HumanEval with DeepSeek-Coder. MGDebugger consistently performs the best across different code lengths.

Figure 8: Repair success rate of different methods when debugging code of different lengths in MBPP with DeepSeek-Coder. MGDebugger consistently performs the best across different code lengths.

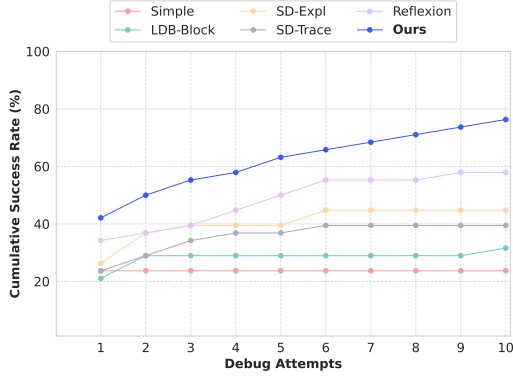

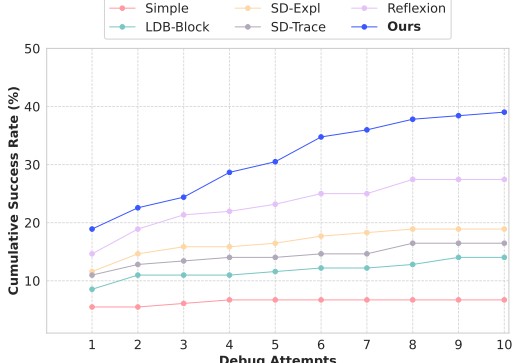

Figure 9: Impact of debug attempts on the cumulative repair success rate of MGDebugger and other methods on HumanEval with DeepSeek-Coder. MGDebugger continues to improve with more debug attempts and outperforms other methods from the beginning to the end.

Figure 10: Impact of debug attempts on the cumulative repair success rate of MGDebugger and other methods on MBPP with DeepSeek-Coder. MGDebugger continues to improve with more debug attempts and outperforms other methods from the beginning to the end.

## F EXAMPLES

We provide example code repairs for HumanEval, MBPP, and HumanEvalFix with DeepSeek-Coder-V2-Lite as the base model. The results of MGDebugger and baselines: Simple Feedback, Self-Edit, LDB (Block), LDB (Line), LDB (Function), Self-Debugging (Expl.), Self-Debugging (Trace) and Reflexion, are shown in the following tables. The buggy part in the original code is highlighted in yellow, and the repaired code is compared with the original buggy code, with changes highlighted in green if the repair passes the final test cases and in red if it fails. The functional comments in the solution code have been replaced with placeholders for brevity, as they are the same as those in the problem description.

The success of code repair often depends on initial solutions, as other methods typically change only a few lines of the original code, keeping the overall structure the same. This tendency to keep the structure of the initial solution may cause other methods to miss important parts of the code that are actually flawed. By contrast, by breaking down the code into smaller parts and looking at different levels of detail, our approach makes these seemingly correct but actually wrong parts easier to spot, as seen in the example of HumanEval. Also, the core of the code is often found in just a few lines,

especially in math or dynamic programming problems. These sections usually need several steps of reasoning to understand and debug. By decomposing them into smaller subfunctions and simulating the execution to trace variable changes, MGDebugger can effectively identify and correct these bugs, as seen in the MBPP and HumanEvalFix examples.

## F.1 EXAMPLE ON HUMANEVAL

**Info:** `HumanEval/54`

**Problem:**
```python
def same_chars(s0: str, s1: str):
    """
    Check if two words have the same characters.
    >>> same_chars('eabcdzzzz', 'dddzzzzzzzddeddabc')
    True
    >>> same_chars('abcd', 'dddddddabc')
    True
    >>> same_chars('dddddddabc', 'abcd')
    True
    >>> same_chars('eabcd', 'dddddddabc')
    False
    >>> same_chars('abcd', 'dddddddabce')
    False
    >>> same_chars('eabcdzzzz', 'dddzzzzzzzddddabc')
    False
    """
```

**Buggy Seed Solution:**
```python
def same_chars(s0: str, s1: str):
    """
    Check if two words have the same characters.
    ...
    """
    return sorted(s0) == sorted(s1)
```

### F.1.1 SIMPLE FEEDBACK

```python
def same_chars(s0: str, s1: str):
    """
    Check if two words have the same characters.
    ...
    """
    return Counter(s0) == Counter(s1)
```

**Result:** ✗

### F.1.2 SELF-EDIT

```python
def same_chars(s0: str, s1: str):
    """
    Check if two words have the same characters.
    ...
    """
    return Counter(s0) == Counter(s1)
```

**Result:** ✗

### F.1.3 LDB (BLOCK)

```python
def same_chars(s0: str, s1: str):
    """
    Check if two words have the same characters.
    ...
    """
    if len(s0) != len(s1):
        return False
```

```
    for char in set(s0):
        if s0.count(char) != s1.count(char):
            return False

    return True
```
**Result:** ✗

### F.1.4  LDB (LINE)

```python
def same_chars(s0: str, s1: str):
    """
    Check if two words have the same characters.
    ...
    """
    return Counter(s0) == Counter(s1)
```
**Result:** ✗

### F.1.5  LDB (FUNCTION)

```python
def same_chars(s0: str, s1: str):
    """
    Check if two words have the same characters.
    ...
    """
    return Counter(s0) == Counter(s1)
```
**Result:** ✗

### F.1.6  SELF-DEBUGGING (EXPL.)

```python
def same_chars(s0: str, s1: str):
    """
    Check if two words have the same characters.
    ...
    """
    return sorted(s0) == sorted(s1)
```
**Result:** ✗

### F.1.7  SELF-DEBUGGING (TRACE)

```python
def same_chars(s0: str, s1: str):
    """
    Check if two words have the same characters.
    ...
    """
    return sorted(s0) == sorted(s1)
```
**Result:** ✗

### F.1.8  REFLEXION

```python
def same_chars(s0: str, s1: str):
    """
    Check if two words have the same characters.
    ...
    """
    if len(s0) > len(s1):
        s0, s1 = s1, s0
    it = iter(s1)
    return all(char in it for char in s0)
```
**Result:** ✗

### F.1.9  MGDEBUGGER

```
1026   def same_chars(s0: str, s1: str) -> bool:
1027       """
1028       Check if two words have the same characters.
1029       ...
1030       """
1031       return sorted_chars_equal(s0, s1)
1032   def sorted_chars_equal(s0: str, s1: str) -> bool:
1033       """Helper function to check if the sorted characters of two strings
1034       are equal."""
1035       return sort_string(s0) == sort_string(s1)
1036   def sort_string(s: str) -> list:
1037       """Helper function to sort the characters of a string."""
1038       return sorted(set(s))
1039
```

**Result:** ✓

## F.2   EXAMPLE ON MBPP

**Info:** `MBPP/185`

**Problem:**
```
def parabola_focus(a, b, c):
    """
    Write a function to find the focus of a parabola.
    """
```

**Buggy Seed Solution:**
```
def parabola_focus(a, b, c):
    """
    Write a function to find the focus of a parabola.
    """
    x_focus = -b / (2 * a)
    y_focus = (1 - (b**2) / (4 * a * c))
    return (x_focus, y_focus)
```

### F.2.1   SIMPLE FEEDBACK

```
def parabola_focus(a, b, c):
    """
    Write a function to find the focus of a parabola.
    """
    x_focus = -b / (2 * a)
    y_focus = c - (b**2) / (4 * a)
    return (x_focus, y_focus)
```

**Result:** ✗

### F.2.2   SELF-EDIT

```
def parabola_focus(a, b, c):
    """
    Write a function to find the focus of a parabola.
    """
    x_focus = -b / (2 * a)
    y_focus = c - (b**2) / (4 * a)
    return (x_focus, y_focus)
```

**Result:** ✗

### F.2.3   LDB (BLOCK)

```
def parabola_focus(a, b, c):
    """
    Write a function to find the focus of a parabola.
    """
```

```
x_focus = -b / (2 * a)
y_focus = (1 - (b**2) / (4 * a * c))
return (x_focus, y_focus)
```

**Result:** ✗

### F.2.4   LDB (LINE)

```
def parabola_focus(a, b, c):
    """
    Write a function to find the focus of a parabola.
    """
    x_focus = -b / (2 * a)
    y_focus = c - (b**2) / (4 * a)
    return (x_focus, y_focus)
```

**Result:** ✗

### F.2.5   LDB (FUNCTION)

```
def parabola_focus(a, b, c):
    """
    Write a function to find the focus of a parabola.
    """
    x_focus = -b / (2 * a)
    y_focus = (1 - (b**2) / (4 * a * c)) # Corrected formula for y_focus
    return (x_focus, y_focus)
```

**Result:** ✗

### F.2.6   SELF-DEBUGGING (EXPL.)

```
def parabola_focus(a, b, c):
    """
    Write a function to find the focus of a parabola.
    """
    x_focus = -b / (2 * a)
    y_focus = c - (b**2) / (4 * a)
    return (x_focus, y_focus)
```

**Result:** ✗

### F.2.7   SELF-DEBUGGING (TRACE)

```
def parabola_focus(a, b, c):
    """
    Write a function to find the focus of a parabola.
    """
    # Calculate the discriminant
    discriminant = b**2 - 4*a*c

    # Calculate the x-coordinate of the focus
    x_focus = -b / (2 * a)

    # Calculate the y-coordinate of the focus
    y_focus = (1 - discriminant) / (4 * a)

    return (x_focus, y_focus)
```

**Result:** ✓

### F.2.8   REFLEXION

```
def parabola_focus(a, b, c):
    """
    Write a function to find the focus of a parabola.
    """
    x_focus = -b / (2 * a)
```

```
y_focus = c - (b**2) / (4 * a)
return (x_focus, round(y_focus, 2)
```

**Result:** ✗

### F.2.9 MGDEBUGGER

```
def parabola_focus(a, b, c):
    """Write a function to find the focus of a parabola."""
    x_focus = calculate_x_focus(a, b)
    y_focus = calculate_y_focus(a, b, c)
    return (x_focus, y_focus)

def calculate_x_focus(a, b):
    """Calculate the x-coordinate of the focus of a parabola."""
    return -b / (2 * a)

def calculate_y_focus(a, b, c):
    """Calculate the y-coordinate of the focus of a parabola."""
    x_vertex = -b / (2 * a)
    y_vertex = a * (x_vertex ** 2) + b * x_vertex + c
    y_focus = y_vertex + 1 / (4 * a)

    return y_focus
```

**Result:** ✓

### F.3 EXAMPLE ON HUMANEVALFIX

**Info:** HumanEval/130

**Problem:**
```
def tri(n):
    """Everyone knows Fibonacci sequence, it was studied deeply by
    mathematicians in
    the last couple centuries. However, what people don't know is
    Tribonacci sequence.
    Tribonacci sequence is defined by the recurrence:
    tri(1) = 3
    tri(n) = 1 + n / 2, if n is even.
    tri(n) =  tri(n - 1) + tri(n - 2) + tri(n + 1), if n is odd.
    For example:
    tri(2) = 1 + (2 / 2) = 2
    tri(4) = 3
    tri(3) = tri(2) + tri(1) + tri(4)
           = 2 + 3 + 3 = 8
    You are given a non-negative integer number n, you have to a return a
     list of the
    first n + 1 numbers of the Tribonacci sequence.
    Examples:
    tri(3) = [1, 3, 2, 8]
    """
```

**Buggy Seed Solution:**
```
def tri(n):
    """

    ...
    """
    if n == 0:
        return [1]
    my_tri = [1, 3]
    for i in range(2, n + 1):
        if i % 2 == 0:
            my_tri.append(i / 2 + 1)
        else:
            my_tri.append(my_tri[i - 1] + my_tri[i - 2] + i + (i + 3) / 2)
```

```
    return my_tri
```

### F.3.1   SIMPLE FEEDBACK

```
def tri(n):
    """
    ...
    """
    if n == 0:
        return [1]
    my_tri = [1, 3]
    for i in range(2, n + 1):
        if i % 2 == 0:
            my_tri.append(1 + i / 2)
        else:
            my_tri.append(my_tri[i - 1] + my_tri[i - 2] + (i + 1) / 2)
    return my_tri
```

**Result:** ✗

### F.3.2   SELF-EDIT

```
def tri(n):
    """
    ...
    """
    if n == 0:
        return [1]
    if n == 1:
        return [1, 3]
    my_tri = [1, 3]
    for i in range(2, n + 1):
        if i % 2 == 0:
            my_tri.append(1 + i // 2) # Use integer division
        else:
            my_tri.append(my_tri[i - 1] + my_tri[i - 2] + my_tri[i - 3])
    return my_tri
```

**Result:** ✗

### F.3.3   LDB (BLOCK)

```
def tri(n):
    """
    ...
    """
    if n == 0:
        return [1]
    my_tri = [1, 3]
    for i in range(2, n + 1):
        if i % 2 == 0:
            my_tri.append(1 + i / 2)
        else:
            my_tri.append(my_tri[i - 1] + my_tri[i - 2] + i + 1) / 2)
    return my_tri
```

**Result:** ✗

### F.3.4   LDB (LINE)

```
def tri(n):
    """
    ...
    """
    if n == 0:
        return [1]
    my_tri = [1, 3, 2] # Initialize with the first three known values
```

```
for i in range(3, n + 1):
    if i % 2 == 0:
        my_tri.append(1 + i / 2)
    else:
        my_tri.append(my_tri[i - 1] + my_tri[i - 2] + my_tri[i - 3])
return my_tri
```

**Result: ✗**

### F.3.5 LDB (FUNCTION)

```
def tri(n):
    """
    ...
    """
    if n == 0:
        return [1]
    my_tri = [1, 3, 2] # Initialize with the first three known values
    for i in range(3, n + 1):
        if i % 2 == 0:
            my_tri.append(1 + i / 2)
        else:
            my_tri.append(my_tri[i - 1] + my_tri[i - 2] + my_tri[i - 3])
    return my_tri
```

**Result: ✗**

### F.3.6 SELF-DEBUGGING (EXPL.)

```
def tri(n):
    """
    ...
    """
    if n == 0:
        return [1]
    if n == 1:
        return [1, 3]
    my_tri = [1, 3]
    for i in range(2, n + 1):
        if i % 2 == 0:
            my_tri.append(1 + i // 2) # Use integer division
        else:
            my_tri.append(my_tri[i - 1] + my_tri[i - 2] + my_tri[i - 3])
    return my_tri
```

**Result: ✗**

### F.3.7 SELF-DEBUGGING (TRACE)

```
def tri(n):
    """
    ...
    """
    if n == 0:
        return [1]
    my_tri = [1, 3, 2] # Initialize with tri(0), tri(1), and tri(2)
    for i in range(3, n + 1):
        if i % 2 == 0:
            my_tri.append(1 + i // 2) # Use integer division
        else:
            my_tri.append(my_tri[i - 1] + my_tri[i - 2] + my_tri[i - 3])
    return my_tri
```

**Result: ✗**

### F.3.8 REFLEXION

```
def tri(n):
    """

    ...
    """
    if n == 0:
        return [1]
    elif n == 1:
        return [1, 3]

    trib_sequence = [1, 3, 2])

    for i in range(3, n + 1):
        if i % 2 == 0:
            trib_sequence.append(1 + i // 2)
        else:
            trib_sequence.append(trib_sequence[i - 1] + trib_sequence[i -
    2] + trib_sequence[i - 3])

    return trib_sequence
```

**Result:** ✗

### F.3.9   MGDEBUGGER

```
def tri(n):
    """

    ...
    """
    if n == 0:
        return [1]
    my_tri = [1, 3]
    for i in range(2, n + 1):
        if i % 2 == 0:
            my_tri.append((1 + i / 2))
        else:
            my_tri.append(tri_odd(my_tri, i))
    return my_tri

def tri_odd(my_tri, i):
    """Helper function to compute the Tribonacci sequence value for odd
    n."""
    if i < 3:
        return None # or return a default value
    return my_tri[i - 1] + my_tri[i - 2] + tri_special(i)

def tri_special(n):
    """Helper function to compute the special value for odd n."""
    return (n + 3) // 2
```

**Result:** ✓

## G   PROMPT TEMPLATES FOR MGDEBUGGER

We provide prompt templates for the three main components of MGDebugger: hierarchical decomposition, test case generation, and sub-function debugging with LLM-simulated execution. These prompts are designed to guide the language model in generating the desired outputs for each step of the debugging process. They have been slightly modified for clarity and brevity, please refer to our codes if you need the exact prompt templates [8].

---

[8]https://anonymous.4open.science/r/MGDebugger-B388

### G.1 PROMPT FOR HIERARCHICAL DECOMPOSITION

> **Prompt Template for Hierarchical Decomposition**
>
> SYSTEM_PROMPT:
> You are an AI assistant specialized in refactoring Python code into a tree-style hierarchical structure.
> USER_PROMPT:
> Convert the following Python code into a tree-style hierarchical structure with multiple levels of sub-functions. Each significant step or logical block should be its own function, and functions can call other sub-functions. Ensure that the main function calls these sub-functions in the correct order, creating a tree-like structure.
> Original Code:
> {code}
> Instruction:
> Please first analyze the codes step by step, and then provide the converted code in a Python code block. When providing the final converted code, make sure to include all the functions in a flattened format, where each function is defined separately.

### G.2 PROMPT FOR TEST CASE GENERATION

> **Prompt for Test Case Generation**
>
> SYSTEM_PROMPT:
> You are an AI assistant specialized in analyzing Python functions and generating test cases.
> USER_PROMPT:
> Full Code:
> {full_code}
> Public Test Cases for the Main Function:
> {public_test_cases}
> Instruction:
> Please analyze how the {function_name} function is used within the main function and how it contributes to the expected outputs in the gold test cases. For each test case, you should analyze step-by-step based on both the input and the expected output of the main function, and then provide the corresponding input and expected output for the {function_name} function. Ensure that the generated test cases are consistent with the behavior expected in the public test cases.

### G.3 PROMPT FOR DEBUGGING SUBFUNCTION

> **Prompt for Debugging Subfunction**
>
> SYSTEM_PROMPT:
> You are an AI assistant helping to debug Python functions.
> USER_PROMPT:
> Debug the following Python function. The function is not passing all test cases. Analyze the code, identify the bug, and provide a fixed version of the function.
> Function Code:
> {function_code}
> Test Case Results:
> {test_case_results}
> Instruction:
> Please try to work as a Python interpreter to execute the code step-by-step. Identify the change of each variable as you "run" the code line-by-line. Based on the execution trace, try to identify the bug and provide the final fixed code in a Python code block.

