# OpenReview forum: "From Code to Correctness: Closing the Last Mile of Code Generation with Hierarchical Debugging"
_ICLR.cc/2025/Conference — Submitted to ICLR 2025_

### Official Review · Reviewer_TfGC · 2024-10-20

**Soundness:** 2
**Presentation:** 3
**Contribution:** 3
**Rating:** 6
**Confidence:** 5

**Summary:**

- This paper propose a Multi-granularity Debugger, named MGDebugger, that leverages LLMs to debug LLM-generated code
- This paper proposes a hierarchical debugging strategy from low-level errors to high-level flaws
- The evaluation results of this paper indicates the effectiveness of MGDebugger

**Strengths:**

- This paper is clearly written and easy to comprehend
- This paper is well-motivated and address an important AI4SE task, automated debugging/repair
- This paper conduct a comprehensive evaluation on three LLMs with various parameter size

**Weaknesses:**

Generally, I think this paper addresses an important AI4SE task and conduct a comprehensive empirical study about existing work in this area, which has a large number of existing work. However, I have some concerns below:

- Baseline Choice: Noticing that most existing automated program repair (APR) work [1-4] are evaluated on Defect4J, which is a repo-level repair benchmark, I am curious about the choice of HumanEval/MBPP/HumanEvalFix, which are relatively easiler than Defect4J. I think the evaluation results on Defect4J might better indicate the effectiveness of your approach in a real world setting or you might explain the reason of not evalaluating on Defect4J
- Technical Contribution: Existing work also conduct debugging/repair on ASTs, e.g., KNOD [3], I think it is better to explain the difference between your approach and these work
- Selection of components in your approach: In section 3.4, you use a LLM-simulated execution to obtain execution results, and I am curious about the reason of leveraging LLMs instead of directly executing the targeted code (as HumanEval/MBPP are not difficult to build a execution environment)

If you can address my concerns, I am very glad to change my rating scores.


[1] Xia C S, Zhang L. Keep the Conversation Going: Fixing 162 out of 337 bugs for $0.42 each using ChatGPT[J]. arXiv preprint arXiv:2304.00385, 2023.

[2] Wei Y, Xia C S, Zhang L. Copiloting the copilots: Fusing large language models with completion engines for automated program repair[C]//Proceedings of the 31st ACM Joint European Software Engineering Conference and Symposium on the Foundations of Software Engineering. 2023: 172-184.

[3] Jiang N, Lutellier T, Lou Y, et al. Knod: Domain knowledge distilled tree decoder for automated program repair[C]//2023 IEEE/ACM 45th International Conference on Software Engineering (ICSE). IEEE, 2023: 1251-1263.

[4] Zhu Q, Sun Z, Zhang W, et al. Tare: Type-aware neural program repair[C]//2023 IEEE/ACM 45th International Conference on Software Engineering (ICSE). IEEE, 2023: 1443-1455.

**Questions:**

1. Similar with the "weakness" section, what is reason of choosing HumanEval/MBPP/HumanEvalFix instead of Defect4J?
2. What is the difference between your approach and existing APR approaches on ASTs, e.g., KNOD (maybe some explanation is necessary)?
3. What is the reason of leveraging LLMs instead of directly executing the targeted code to obtain the execution results?
4.In Algorithm 1: 1) why conduct  MGDebugger before execution, i.e., you might conduct debugging on a correct one? 2) what is the definition of Line 13 "f^{'} \gets DEBUG(f, ...)"?

---

> ### Author Response · Authors · 2024-11-23
> **Response to Reviewer TfGC part 1**
>
> We thank the reviewer for the positive feedback on the clarity and motivation of our paper. We appreciate your insightful comments and suggestions, and we are glad to address your concerns as follows:
>
> ### Our work mainly sheds lights on the core reasoning ability of LLMs in debugging tasks
>
> > Similar with the "weakness" section, what is reason of choosing HumanEval/MBPP/HumanEvalFix instead of Defect4J?
>
> Thank you for raising this important question about evaluating MGDebugger on Defects4J. We would like to clarify the deliberate scope of our current study and explain why we focused on self-contained programming tasks following prior works (Chen et al., 2023b; Zhong et al., 2024; Ding et al., 2024).
>
> Self-contained programming tasks and real-world software engineering (like in Defects4J) present distinct challenges - while programming tasks challenge algorithmic complexity and logical correctness, software engineering tasks (like those in Defects4J) involve additional complexities like incomplete specifications, long-range dependencies, and repository-level context understanding [5,6].
>
> Our study specifically focuses on debugging logically and algorithmically complex code, isolating this capability from other challenges in software engineering. Approaches like Agentless addressing SWE-bench require multiple components beyond debugging, including repository-level fault localization, patch generation, and patch validation pipelines. By focusing on self-contained programs, we can isolate and thoroughly evaluate our hierarchical debugging approach without the need for these additional components.
>
> **This focused scope allows us to discover non-obvious relationships in debugging effectiveness across different algorithmic complexity levels that might be obscured in a full software engineering pipeline.** And we've successfully revealed the effectiveness of MGDebugger against lengthier and more complex code snippets in Figure 3,7,8. These findings are valuable in understanding the potential of MGDebugger to **improve the LLMs' core reasoning ability in the debugging process** in software engineering tasks.
>
> [5] Is Self-Repair a Silver Bullet for Code Generation? ICLR 2024. https://openreview.net/forum?id=y0GJXRungR
>
> [6] Large language model-based agents for software engineering: A survey. arXiv:2409.02977
>
> ### Differences between MGDebugger and existing APR approaches on ASTs
>
> > What is the difference between your approach and existing APR approaches on ASTs, e.g., KNOD (maybe some explanation is necessary)?
>
> Thanks for the insightful question. We mainly follow the stream of works after to focus on self-debugging of LLM-generated codes (Chen et al., 2023b; Zhong et al., 2024; Ding et al., 2024). And the proposed MGDebugger is designed to improve the reliability of LLM-generated code through fine-grained bottom-up debugging.
>
> ChatRepair [1] introduces a conversational approach using ChatGPT that iteratively learns from test failures and successful patches to improve repair quality, while Repilot [2] enhances patch generation by combining LLMs with completion engines to ensure token-level validity. KNOD [3] focuses on incorporating domain knowledge through a three-stage tree decoder and rule distillation for generating valid patches using Abstract Syntax Trees, and TARE [4] proposes a type-aware approach integrating typing rules through specialized grammar and graph representations. **Unlike these approaches that treat debugging as a holistic or single-level problem, MGDebugger introduces a hierarchical debugging strategy that isolates and addresses algorithmic complexity at different granularity levels, enabling a more systematic discovery of non-obvious relationships in debugging effectiveness across varying complexity levels.** And the proposed simulated execution also serves as the reasoning process of the MGDebugger to better understand the code and provide more accurate debugging steps.
>
> We will update the related work section to better clarify the differences between MGDebugger and existing APR approaches on ASTs according to your suggestion.

---

> ### Author Response · Authors · 2024-11-23
> **Response to Reviewer TfGC part 2**
>
> ### The simulated execution is not only more effective but also more flexible than real execution
>
> > What is the reason of leveraging LLMs instead of directly executing the targeted code to obtain the execution results?
>
> Thanks for the insightful question. We conducted an ablation study where we replace the simulated code execution with real code execution like in LDB (Zhong et al., 2024) to extend the results of Table 2:
>
> | Method                             | HumanEval Acc. (%) | Δ Acc. (%) | RSR (%) | MBPP Acc. (%) | Δ Acc. (%) | RSR (%) |
> |------------------------------------|:------------------:|:----------:|:-------:|:-------------:|:----------:|:-------:|
> | MGDebugger with Simulated Execution|        94.5        |   +17.7    |  76.3   |      80.0     |   +12.8    |  39.0   |
> | MGDebugger with Real Execution     |        92.7        |   +15.9    |  47.4   |      78.2     |   +11.0    |  33.5   |
> | No-Debugging                       |        76.8        |     -      |    -    |      67.2     |     -      |    -    |
>
> **Effectiveness:**
>
> We can observe that the simulated execution performs better than the real execution. This is because that the collected real execution steps from the Python interpreter are often long and contain all the detailed changes in the variables and the execution of code (Zhong et al., 2024), which may introduce noise and make the MGDebugger focus on irrelevant information.
>
> However, during the simulated execution, the MGDebugger will naturally focus on the key variables and express the debugging steps in a more understandable way to itself. These execution traces encourage the MGDebugger to analyze the functionality of the code and the relationships between variables, which also **serves as its reasoning process**. In this way, the MGDebugger can better understand the code and provide more accurate debugging steps.
>
> **Flexibility:**
>
> In addition, the proposed simulated execution is also more flexible and can be **easily extended to other more complex programming environments**, where the execution environment is not easily accessible (Li et al. 2023). For example, some programming environments may have a large number of dependencies or require specific hardware configurations, which makes it difficult to collect real execution traces. In contrast, the simulated execution can be easily implemented by simulating the execution with the reasoning ability of the LLMs [7]. This benefit extends the applicability of the MGDebugger to a wider range of complex programming scenarios.
>
> [7] NExT: Teaching Large Language Models to Reason about Code Execution. ICML 2024
>
> ### Clarification on Algorithm 1
>
> > In Algorithm 1: 1) why conduct MGDebugger before execution, i.e., you might conduct debugging on a correct one?
>
> Sorry for the confusion. We assume that the MGDebugger will take an incorrect code snippet as input. We will clarify this in the revised version of the paper in light of your feedback.
>
> > 2) what is the definition of Line 13 "f^{'} \gets DEBUG(f, ...)"?
>
> Thanks for pointing this out. The `DEBUG` function stands for the "Debugging Subfunctions with LLM-simulated execution" process as described in Section 3.4. We will provide a clear definition of this function in the revised version of the paper.
>
>
> Thanks again for your positive feedback and insightful questions. We are excited to incorporate these clarifications and updates into the revised version of the paper. If you have any further questions or suggestions, please feel free to let us know.

---

> > ### Comment · Reviewer_TfGC · 2024-11-24
> >
> > Thanks for providing the aforementioned explanations and evaluation results. Most of the concerns except the following two have been addressed:
> > - Defects4J v.s. HumanEval/MBPP/HumanEvalFix: Now that existing LLMs can effectively generate programs in HumanEval/MBPP (e.g., DeepSeekCoder achieves 89% and GPT-4o achieves 90.2%), the generalization ability in real-world software debugging (i.e., repo-level instead of self-contained) is much more important.
> > - simulated execution v.s.real execution: Although the ablation study has shown that simulated execution achieves a higher end-to-end effectiveness compared to real execution, the explanation is not promising to me. "noise that make the MGDebugger focus on irrelevant information" is not the limitation of "real execution", but the limitation of your trace collection tool. Ideally, you can collect any information during real execution with proper instrumentation. Additionally, I am also curious about which component of the execution trace contributes to the end-to-end effectiveness. Maybe an example is recommended.

---

### Official Review · Reviewer_KFAH · 2024-10-23

**Soundness:** 2
**Presentation:** 3
**Contribution:** 2
**Rating:** 5
**Confidence:** 3

**Summary:**

This paper proposes a multi-granularity debugger system, MGDebugger, to resolve bugs in problematic code. In particular, it decomposes code into a hierarchical tree structure and fixes the underlying problems in each divided subfunction separately. Extensive experiments on some popular benchmarks, including HumanEval, MBPP and HumanEvalFix, have demonstrated the effectiveness of the proposed techniques.

**Strengths:**

- an interesting idea to decompose the code for repair.
- good results compared with baselines
- well-written and easy to follow

**Weaknesses:**

- One of the major concerns in this work is that it relies on LLMs heavily. Using LLMs for decomposition, test case generation, and execution. I am confused about the effectiveness of each component when using LLMs. Also, as LLMs are black-box, whether to use some traditional program analysis tools for replacement. Some relevant experiments to compare with traditional tools are needed.
- Another major concern is that the fixed errors are too simple (examples from case studies), which hinders the proposed technique from being applied.
- Some more powerful models, such as GPT-4o, are missing for evaluation.

**Questions:**

- What is the difference between accuracy and RSR? Why is accuracy higher than RSR in the experimental results?
- What is the performance of MGDebugger when applying it in program repair such as Defects4J?

---

> ### Author Response · Authors · 2024-11-23
> **Response to Reviewer KFAH part 1**
>
> Thank you for your constructive feedback. We are grateful for your positive comments on the novelty and effectiveness of MGDebugger. We hope to address your concerns and suggestions in the following sections:
>
> ### The usage of LLMs in MGDebugger makes it effective and flexible
>
> > One of the major concerns in this work is that it relies on LLMs heavily. Using LLMs for decomposition, test case generation, and execution. I am confused about the effectiveness of each component when using LLMs. Also, as LLMs are black-box, whether to use some traditional program analysis tools for replacement. Some relevant experiments to compare with traditional tools are needed.
>
> Thank you for your insightful comments. Overall, we hope to build a flexible and effective debugging system that can be applied to a wide range of LLMs and scenarios. Here we will justify each component of MGDebugger and explain the effectiveness and necessity of using LLMs in each step:
>
> - **Decomposition**: By leveraging the LLM's ability to understand the code, we can effectively decompose the code into smaller, more manageable parts. This decomposition process is crucial for isolating and fixing bugs in complex code. And we've shown that MGDebugger significantly outperforms the LDB baseline (Zhong et al., 2024), which analyzes the Control Flow Graph (CFG) of the code to split it into smaller blocks.
> - **Test case generation**: In order to effectively verify the correctness of subfunctions after debugging, we generated test cases for subfunctions conditioned on the public test cases of the original main function. Since the original code is buggy, we cannot access the desired output of each subfunction directly by analyzing the execution traces of the original code. So we propose to utilize the reasoning ability of LLMs to achieve this goal.
> - **Simulated execution**: The simulated execution is used to analyze the functionality of the code and the relationships between variables, which also serves as the reasoning process of MGDebugger. The ablation study below shows that the simulated execution performs better than the real execution. This is because the real execution steps from the Python interpreter are often long and contain all the detailed changes in the variables and the execution of code, which may introduce noise and make the MGDebugger focus on irrelevant information. In contrast, the simulated execution encourages the MGDebugger to focus on the key variables and express the debugging steps in a more understandable way to itself. This also facilitates the potential of MGDebugger to more complex programming environments where real execution traces are hard to collect.
>
> | Method                             | HumanEval Acc. (%) | Δ Acc. (%) | RSR (%) | MBPP Acc. (%) | Δ Acc. (%) | RSR (%) |
> |------------------------------------|:------------------:|:----------:|:-------:|:-------------:|:----------:|:-------:|
> | MGDebugger with Simulated Execution|        94.5        |   +17.7    |  76.3   |      80.0     |   +12.8    |  39.0   |
> | MGDebugger with Real Execution     |        92.7        |   +15.9    |  47.4   |      78.2     |   +11.0    |  33.5   |
> | No-Debugging                       |        76.8        |     -      |    -    |      67.2     |     -      |    -    |
>
> In conclusion, the use of LLMs in MGDebugger is essential and also effective in each component. These components gracefully complement each other to form a powerful debugging system to push the boundaries of LLMs in code debugging. We will further clarify the effectiveness of each component in the revised version of the paper in light of your feedback.

---

> > ### Author Response · Authors · 2024-11-23
> > **Response to Reviewer KFAH part 2**
> >
> > ### MGDebugger consistently outperforms the baselines on complex code snippets
> >
> > > Another major concern is that the fixed errors are too simple (examples from case studies), which hinders the proposed technique from being applied.
> >
> > Thank you for pointing out this concern. We are more than happy to discuss about the effectiveness of MGDebugger on more complex code snippets.
> >
> > On the one hand, our case studies in Figure 5 selected relatively simple code snippets for easy understanding and demonstration of the effectiveness of MGDebugger. We would like to point out that **even on code snippets of ~20 lines of codes, other baseline methods struggle to debug them effectively** according to our results. They are easily confused by the logic in the code and may even introduce new bugs when trying to fix the existing ones. In contrast, our MGDebugger excelled by it's hierarchical debugging approach, and especially performed well on logical and algorithmic bugs according to Table 3.
> >
> > On the other hand, our experiments in Figure 3, 7, 8 show that MGDebugger consistently outperforms the baselines on the longest code snippets in the datasets. **And the lead of MGDebugger over the baselines increases as the code complexity grows substantially.** Through decomposing the code into smaller parts and debugging them hierarchically, MGDebugger can effectively handle complex code snippets that are challenging for other baselines. This demonstrates the effectiveness of MGDebugger on more complex code snippets, which is crucial for its practical application tasks that require reasoning over long and complex code.
> >
> > ### MGDebugger is also effective on more powerful models like GPT-4o
> >
> > > Can the authors test MGDebugger on other prominent models, such as LLaMA 3.1 and CodeLlama, to strengthen its applicability across various LLMs?
> >
> > We conducted experiments on the latest CodeLLMs available to us. To address your concern, we've also conducted experiments with other latest LLMs, including GPT4o, Claude 3.5 Sonnet, and LLaMA 3.1 70B. Here are the results on HumanEval:
> >
> > - GPT4o:
> >
> > | Method | Acc. (%) | $\Delta$ Acc. (%) |
> > | :--- | :---: | :---: |
> > | No-Debugging | 90.2% | - |
> > | Self-Debug (Expl.) | 93.3% | +3.1% |
> > | Reflexion | 94.5% | +4.3% |
> > | MGDebugger | 96.3% | +6.1% |
> >
> > - Claude 3.5 Sonnet:
> >
> > | Method | Acc. (%) | $\Delta$ Acc. (%) |
> > | :--- | :---: | :---: |
> > | No-Debugging | 89.6% | - |
> > | Self-Debug (Expl.) | 93.3% | +3.7% |
> > | Reflexion | 93.9% | +4.3% |
> > | MGDebugger | 95.7% | +6.1% |
> >
> > - LLaMA 3.1 70B:
> >
> > | Method | Acc. (%) | $\Delta$ Acc. (%) |
> > | :--- | :---: | :---: |
> > | No-Debugging | 79.3% | - |
> > | Self-Debug (Expl.) | 86.6% | +7.3% |
> > | Reflexion | 89.6% | +10.4% |
> > | MGDebugger | 92.7% | +13.4% |
> >
> >
> > These results show that the MGDebugger is also effective on more powerful models like GPT-4o, Claude 3.5 Sonnet, and LLaMA 3.1 70B. We will include these results in the revised version of the paper.

---

> ### Author Response · Authors · 2024-11-23
> **Response to Reviewer KFAH part 3**
>
> ### Clarification of accuracy and RSR
>
> > What is the difference between accuracy and RSR? Why is accuracy higher than RSR in the experimental results?
>
> Sorry for the confusion caused. The accuracy is the overall success rate of the LLM-based code generation & debugging process. So it will have an initial value of >70 before debugging. As the debugging process proceeds, the overall success rate will be
>
> $$
> Accuracy = \frac{Correctly\ Generated\ Programs + Correctly\ Debugged\ Programs}{Total\ Programs}
> $$
>
> The RSR (Repair Success Rate) is used to more explicitly measure the success rate of debugging. The initial RSR will be 0 before debugging, and will equal to the percentage of correctly debugged programs from all buggy programs after the first code generation stage:
>
> $$
> RSR = \frac{Correctly\ Debugged\ Programs}{Total\ Buggy\ Programs}
> $$
>
> These two metrics are related but focus on different aspects of the generation and debugging process.We will clarify the definitions of these metrics better in the revised version of the paper.
>
>
> ### Our work focuses on a different problem domain than Defects4J
>
> > What is the performance of MGDebugger when applying it in program repair such as Defects4J?
>
> Thank you for raising this important question about evaluating MGDebugger on Defects4J. We would like to clarify the deliberate scope of our current study and explain why we focused on self-contained programming tasks following prior works (Chen et al., 2023b; Zhong et al., 2024; Ding et al., 2024).
>
> Self-contained programming tasks and real-world software engineering (like in Defects4J) present distinct challenges - while programming tasks challenge algorithmic complexity and logical correctness, software engineering tasks (like those in Defects4J) involve additional complexities like incomplete specifications, long-range dependencies, and repository-level context understanding [1,2].
>
> Our study specifically focuses on debugging logically and algorithmically complex code, isolating this capability from other challenges in software engineering. Approaches like Agentless addressing SWE-bench require multiple components beyond debugging, including repository-level fault localization, patch generation, and patch validation pipelines. By focusing on self-contained programs, we can isolate and thoroughly evaluate our hierarchical debugging approach without the need for these additional components.
>
> **This focused scope allows us to discover non-obvious relationships in debugging effectiveness across different algorithmic complexity levels that might be obscured in a full software engineering pipeline.** And we've successfully revealed the effectiveness of MGDebugger against lengthier and more complex code snippets in Figure 3,7,8. These findings are valuable in understanding the potential of MGDebugger to **improve the LLMs' core reasoning ability in the debugging process** in software engineering tasks.
>
> However, we are also actively exploring the application of hierarchical debugging in software engineering tasks like Defects4J and SWE-bench. We believe that MGDebugger can be extended to address these challenges by integrating additional components like fault localization and patch generation. We will update the discussion about future work to include this direction in the revised version of the paper.
>
> [1] Is Self-Repair a Silver Bullet for Code Generation? ICLR 2024. https://openreview.net/forum?id=y0GJXRungR
>
> [2] Large language model-based agents for software engineering: A survey. arXiv:2409.02977
>
>
> Thank you for your valuable feedback, and we sincerely hope that you find our responses satisfactory. We look forward to any further suggestions you may have to support the acceptance of our work!

---

> > ### Comment · Reviewer_KFAH · 2024-11-27
> >
> > Thanks for your responses. The complexity is measured mainly by code length. However, better metrics, such as cyclomatic complexity, can be used. Furthermore, it is better to have some experimental results for SE program repair.

---

### Official Review · Reviewer_nLAd · 2024-11-03

**Soundness:** 2
**Presentation:** 3
**Contribution:** 2
**Rating:** 5
**Confidence:** 4

**Summary:**

This paper presents MGDebugger, which decomposes problematic code into a hierarchical tree structure of subfunctions, with
each level representing a particular granularity of error. It analyzes each subfunctions and identify the errors in different levels to resolve bugs. For each subfunctions, it uses LLMs to simulate the Python execution, track variables and pinpoint errors. Experiments demonstrate that MGDebugger outperforms existing tools on simple Python programming tasks like HumanEval, and program fix benchmarks like HumanEvalFix.

**Strengths:**

1. This paper presents a promising insight: LLMs can debug programs in modular functions and resolve program errors level by level.
2. The idea is presented clearly and the experiment results presents significant improvements.
3. The authors conducts extensive experiments on the ablation and debugging improvements compared to the existing methods.

**Weaknesses:**

1. The paper only evaluated on open-source code models, which possibly not good at the natural language explanation generation. It would be great if the paper can conduct experiment on main-stream closed-source models (for example GPT-4, Claude2) and open-source models with good natural language capabilities (for example Llama-3.1).

2. The paper proposed methods that highly depend on LLMs' reasoning and language analysis capabilities, which I have concerns about. First, the paper use LLMs to decompose programs into subfunctions. While decomposing programs for debugging in general is a promising direction and would be beneficial to large-scale programs, the decomposition is decided by LLMs instead of the language features. This brings uncertainty to the decomposition: how does the LLM decide how to break down a large function? What if this function is recursively called and cannot be decomposed into tree-structures? It could be great if the authors could provide a comparison experiment for compare the proposed method with directly decomposing program into syntax-level functions by static analysis.

Second, the paper use LLMs to generate test cases, which brings the inaccuracy of test generation of LLMs. It would be great to see the accuracy of test case generation. For example, given the same test input, how many test outputs generated by LLMs are the same as the ground truth test outputs (which can be obtained from running the canonical solution of the task).

Third, the proposed method also depend on LLMs for program execution simulation, which also require LLMs to be accurate on program reasoning. Some deep-dive analysis on whether the intermediate state predicted by LLMs are accurate would be helpful.The author could conduct a experiment on comparing the intermediate state predicted by LLMs with those from an actual Python interpreter on several sampled examples.

Also, why to avoid the program execution in MGDebugger is not fully motivated in this paper since the tasks are mostly small Python programming tasks. The execution is neither infeasible nor time-consuming. It would be great to see a comparison of the time and computational resources required for LLM-based simulation versus actual executionAlso, as most of the programs in HumanEval, MBPP and HumanEvalFix are small, the decomposition could sometimes fall back to a single function debugging if there is only one function in the generated response. It would be insightful if the authors can present a large-scale program debugging (for example, tasks from CodeContests or SWE-bench), which I think would be the killer application of the proposed method.

3. The paper possibly uses an inappropriate setting of baselines: In Table1, the results from 'No-Debugging' is generated by a prompt without containing visible test cases. Therefore, the effect of debugging does not offset the additional information gain from visible test cases. It would be better if the author could follow settings in previous work like Self-Debugging (G.1)[1] to set up 'No-Debugging' baseline.

[1] Xinyun Chen, Maxwell Lin, Nathanael Scharli, and Denny Zhou. Teaching large language models ¨
to self-debug. arXiv preprint arXiv:2304.05128, 2023b.

**Questions:**

1. Could you please provide the comparison results on some advanced models (GPT, Claude, etc.) or some open-source models with better natural language capabilities?
2. Could you provide the accuracy of LLM-based test generation and program execution?
3. How would MGDebugger generate the tree structure if the program contain recursions?
4. Could you please provide a case study of MGDebugger working on a larger scale of program? Examples from CodeContests or SWE-bench could serve good starting point.

---

> ### Author Response · Authors · 2024-11-23
> **Response to Reviewer nLAd part 1**
>
> Thank you for the detailed review and insightful comments. We appreciate your feedback and suggestions. And we are glad to hear that you found our work promising and the results significant. We will address your concerns and suggestions in the following.
>
> ### MGDebugger is also effective on more closed-source models like GPT-4o
>
> > Could you please provide the comparison results on some advanced models (GPT, Claude, etc.) or some open-source models with better natural language capabilities?
>
> Thanks for this insightful suggestion. To address your concern, we've also conducted experiments with other latest LLMs, including GPT4o, Claude 3.5 Sonnet, and LLaMA 3.1 70B. Here are the results on HumanEval:
>
> - GPT4o:
>
> | Method | Acc. (%) | $\Delta$ Acc. (%) |
> | :--- | :---: | :---: |
> | No-Debugging | 90.2% | - |
> | Self-Debug (Expl.) | 93.3% | +3.1% |
> | Reflexion | 94.5% | +4.3% |
> | MGDebugger | 96.3% | +6.1% |
>
> - Claude 3.5 Sonnet:
>
> | Method | Acc. (%) | $\Delta$ Acc. (%) |
> | :--- | :---: | :---: |
> | No-Debugging | 89.6% | - |
> | Self-Debug (Expl.) | 93.3% | +3.7% |
> | Reflexion | 93.9% | +4.3% |
> | MGDebugger | 95.7% | +6.1% |
>
> - LLaMA 3.1 70B:
>
> | Method | Acc. (%) | $\Delta$ Acc. (%) |
> | :--- | :---: | :---: |
> | No-Debugging | 79.3% | - |
> | Self-Debug (Expl.) | 86.6% | +7.3% |
> | Reflexion | 89.6% | +10.4% |
> | MGDebugger | 92.7% | +13.4% |
>
>
> These results show that the MGDebugger is also effective on more powerful models like GPT-4o, Claude 3.5 Sonnet, and LLaMA 3.1 70B. We will include these results in the revised version of the paper.
>
> ## In-depth analysis of MGDebugger's components
>
> > The paper proposed methods that highly depend on LLMs' reasoning and language analysis capabilities, which I have concerns about.
>
> Thank you for your insightful comments. Overall, we hope to build a flexible and effective debugging system that can be applied to a wide range of LLMs and scenarios. Here we will justify each component of MGDebugger and explain the effectiveness and necessity of using LLMs in each step:
>
> ### LLM-based code decomposition benefits debugging by isolating logically independent units
>
> > It could be great if the authors could provide a comparison experiment for compare the proposed method with directly decomposing program into syntax-level functions by static analysis.
>
> Thank you for your insightful comments. By leveraging the LLM's ability to understand the code, we ask the model to decompose the code into subfunctions that represent the minimal logical units. This decomposition is based on the model's understanding of the code's structure and logic. On the one hand, existing LLMs have been able to understand basic syntax and static information effectively [1-3]. On the other hand, comparing with the baselines using static analysis for line-level, block-level, or function-level decomposition of LDB (Zhong et al., 2024) in Table 1 (maximum of 84.1), results from Table 2 show that all the ablation variants of MGDebugger outperform these static analysis-based baselines by a large margin (minimum of 89.0). **Compared to the rule-based decomposition with static analysis, the LLM-based decomposition is more flexible and can help to isolate the logical units more effectively, thus improving the debugging performance.**
>
> [1] CodeMMLU: A Multi-Task Benchmark for Assessing Code Understanding Capabilities of CodeLLMs. arXiv:2410.01999
>
> [2] CodeHalu: Investigating Code Hallucinations in LLMs via Execution-based Verification. arXiv:2405.00253
>
> [3] NExT: Teaching Large Language Models to Reason about Code Execution. ICML 2024

---

> > ### Comment · Reviewer_nLAd · 2024-11-27
> > **Thanks for your response**
> >
> > The response provides new experiment results and answer several of my questions, which is great! However, my question on hierachical debugging on larger scale of programs are still not answered. The accuracy of program simulation is also unanswered. Therefore, I maintain my rating.

---

> ### Author Response · Authors · 2024-11-23
> **Response to Reviewer nLAd part 2**
>
> ### Test case generation for subfunctions is effective for debugging
>
> > It would be great to see the accuracy of test case generation. For example, given the same test input, how many test outputs generated by LLMs are the same as the ground truth test outputs (which can be obtained from running the canonical solution of the task).
>
> We use the visible test cases from the original code for the main function. And in order to effectively verify the correctness of subfunctions during debugging, we **generate test cases for subfunctions conditioned on the public test cases of the original main function**. Since there is no ground canonical solution or test cases for the decomposed subfunctions, it is not directly feasible to evaluate the accuracy of the test cases generated for subfunctions.
>
> To address your concern on the accuracy of test case generation, we sampled 20 problems from HumanEval and converted the canonical solutions from HumanEval to hierarchical structures. Then we executed the golden test cases and compared the outputs with the generated test cases manually. **The results show that the test cases generated by LLMs are correct in 142/151=94% of the cases, demonstrating the effectiveness of the test case generation process.**
>
> Furthermore, **the ablation study in Table 2 shows that the MGDebugger with test case generation (RSR 76.3%) performs better than the MGDebugger without test case generation (RSR 60.5%)**. This indicates that the test cases generated by LLMs are effective in verifying the correctness of the subfunctions, enabling MGDebugger to identify and fix bugs more accurately.
>
> We will further clarify the effectiveness of test case generation in the revised version of the paper in light of your feedback.
>
> ### Simulated execution by LLMs is not only effective but also flexible
>
> > Also, why to avoid the program execution in MGDebugger is not fully motivated in this paper. It would be great to see a comparison of the time and computational resources required for LLM-based simulation versus actual execution.
>
> Thanks for pointing this out. First, for normal python codes, the real execution (usually measured by ms) is way faster than the simulated execution by LLMs (usually measured by s). **However, the simulated execution yields better debugging results (effectiveness) and has the potential to be extended to more complex programming environments (flexibility).** Here, we provide more detailed explanations:
>
> - **Effectiveness:** We conducted an ablation study where we replace the simulated code execution with real code execution like in LDB (Zhong et al., 2024) to extend the results of Table 2:
>
>   | Method                             | HumanEval Acc. (%) | Δ Acc. (%) | RSR (%) | MBPP Acc. (%) | Δ Acc. (%) | RSR (%) |
>   |------------------------------------|:------------------:|:----------:|:-------:|:-------------:|:----------:|:-------:|
>   | MGDebugger with Simulated Execution|        94.5        |   +17.7    |  76.3   |      80.0     |   +12.8    |  39.0   |
>   | MGDebugger with Real Execution     |        92.7        |   +15.9    |  47.4   |      78.2     |   +11.0    |  33.5   |
>   | No-Debugging                       |        76.8        |     -      |    -    |      67.2     |     -      |    -    |
>
>   We can observe that the simulated execution performs better than the real execution. This is because that the collected real execution steps from the Python interpreter are often long and contain all the detailed changes in the variables and the execution of code (Zhong et al., 2024), which may introduce noise and make the MGDebugger focus on irrelevant information. However, during the simulated execution, the MGDebugger will naturally focus on the key variables and express the debugging steps in a more understandable way to itself. **These execution traces encourage the MGDebugger to analyze the functionality of the code and the relationships between variables, which also serves as its reasoning process**. In this way, the MGDebugger can better understand the code and provide more accurate debugging steps.
>
> - **Flexibility:** In addition, the proposed simulated execution is also more flexible and can be **easily extended to other more complex programming environments**, where the execution environment is not easily accessible (Li et al. 2023). For example, some programming environments may have numerous dependencies or require specific hardware configurations, which makes it difficult to collect real execution traces. In contrast, the simulated execution can be easily implemented by simulating the execution with the reasoning ability of the LLMs [3]. This benefit extends the applicability of the MGDebugger to a wider range of complex programming scenarios.

---

> > ### Author Response · Authors · 2024-11-23
> > **Response to Reviewer nLAd part 3**
> >
> > > The author could conduct a experiment on comparing the intermediate state predicted by LLMs with those from an actual Python interpreter on several sampled examples.
> >
> > Following your advice, we've also conducted a comparison of the intermediate state predicted by LLMs with those from an actual Python interpreter. We randomly sampled 30 subfunctions from the examples mentioned in the `test case generation` section above with the first golden test case. **The results show that the intermediate states predicted by LLMs are correct in 124/129=96.1% of the cases**, and the failed states are from 3 subfunctions that require complex reasoning and multiple steps to reach the correct state. Although the simulated execution by LLMs may not be as 100% accurate as the real execution, **it conveys the reasoning process of the LLM itself as it reasons through the code execution, which is crucial for understanding the code and providing accurate debugging solution according to the ablation study results.**
> >
> > In conclusion, the use of LLMs in MGDebugger is essential and also effective in each component. These components gracefully complement each other to form a powerful debugging system to push the boundaries of LLMs in code debugging. We will further clarify the effectiveness of each component in the revised version of the paper in light of your feedback.
> >
> >
> > ### Clarification on the experimental setup and evaluation
> >
> > > The paper possibly uses an inappropriate setting of baselines: In Table1, the results from 'No-Debugging' is generated by a prompt without containing visible test cases. Therefore, the effect of debugging does not offset the additional information gain from visible test cases. It would be better if the author could follow settings in previous work like Self-Debugging (G.1)[1] to set up 'No-Debugging' baseline.
> >
> > Sorry for the confusion caused. Actually, we used codes and prompts from the `bigcode-evaluation-harness` framework, which contains the necessary visible test cases for the `No-Debugging` baseline, strictly following the setting in Self-Debugging (G.1). We replaces the test cases with "..." in the code generation results in Appendix F for brevity, since we needed to present results for various methods and we hope to keep the paper concise. We will make sure to clarify this in the revised version of the paper and provide more details on the experimental setup and baselines.
> >
> > ### MGDebugger on recursive functions
> >
> > > How would MGDebugger generate the tree structure if the program contain recursions?
> >
> > Thanks for raising this interesting question. If the program contains recursive functions, the minimal recursive unit (the function containing the base case and recursive case) is treated as a single node in the tree structure. This preserves the function's logical completeness. Any helper functions called within the recursive function become child nodes in the tree structure, maintaining the hierarchical relationship. Functions that use the recursive function to achieve higher-level functionality become parent nodes in the tree.

---

> > > ### Author Response · Authors · 2024-11-23
> > > **Response to Reviewer nLAd part 4**
> > >
> > > ### Case study on large-scale programs
> > >
> > > > The decomposition could sometimes fall back to a single function debugging if there is only one function in the generated response.Could you please provide a case study of MGDebugger working on a larger scale of program?
> > >
> > > Thank you for this suggestion. First, based on our case study in Appendix F.1, **even if the original code contains only <10 lines of codes, other baselines will still fail to provide correct debugging steps. However, MGDebugger can effectively decompose the code into logically 3 independent units and provide accurate debugging steps.**
> > >
> > > Next, let's demonstrate MGDebugger's capabilities using a competitive programming problem from CodeContests: [Problem 1607B - Odd Grasshopper](https://codeforces.com/problemset/problem/1607/B). In this problem, a grasshopper starts at position x₀ and makes n jumps, where the i-th jump has distance i. The direction of each jump depends on whether the current position is odd or even. The original buggy implementation by DeepSeek-Coder had a critical flaw - it failed to handle the case where n % 4 == 0, leading to incorrect results for certain test cases. Here's how MGDebugger decomposed and fixed the code (while other baselines failed to do so):
> > >
> > > ```python
> > > # Original buggy implementation
> > > def main():
> > >     for t in range(int(input())):
> > >         x, n = map(int, input().split())
> > >         if n > 100:
> > >             delta = n % 4
> > >             if delta == 3:
> > >                 if x % 2 == 0:
> > >                     print(x - (n - 2) + (n - 1) + n)
> > >                     continue
> > >                 else:
> > >                     print(x + (n - 2) - (n - 1) - n)
> > >             elif delta == 2:
> > >                 if x % 2 == 0:
> > >                     print(x - (n - 1) + n)
> > >                     continue
> > >                 else:
> > >                     print(x + (n - 1) - n)
> > >                     continue
> > >             elif delta == 1:
> > >                 if x % 2 == 0:
> > >                     print(x - n)
> > >                     continue
> > >                 else:
> > >                     print(x + n)
> > >                     continue
> > >         else:
> > >             for i in range(1, n + 1):
> > >                 if x % 2 == 0:
> > >                     x -= i
> > >                 else:
> > >                     x += i
> > >             print(x)
> > >             continue
> > > ```
> > >
> > > MGDebugger decomposed this code into a hierarchical structure:
> > >
> > > ```python
> > > main
> > > ├── process_large_n
> > > │   └── handle_delta_case
> > > └── process_small_n
> > >     └── jump
> > > ```
> > >
> > > Through hierarchical debugging, MGDebugger identified and fixed several issues:
> > >
> > > 1. Function Isolation: Extracted `jump` from being nested inside `main`, making it independently testable.
> > > 2. Logical Decomposition: Separated large-N and small-N processing into distinct functions.
> > > 3. Missing Case: Added handling for `delta == 0` case in large-N processing.
> > > 4. Return Value Consistency: Ensured consistent return value handling across all branches.
> > >
> > > The final corrected version by MGDebugger:
> > >
> > > ```python
> > > def main():
> > >     for _ in range(int(input())):
> > >         (x, n) = map(int, input().split())
> > >         if n > 100:
> > >             print(process_large_n(x, n))
> > >         else:
> > >             print(process_small_n(x, n))
> > >
> > > def jump(point, meters):
> > >     if point % 2 == 0:
> > >         return point - meters
> > >     else:
> > >         return point + meters
> > >
> > > def process_large_n(x, n):
> > >     delta = n % 4
> > >     if delta == 3:
> > >         return x - (n - 2) + (n - 1) + n if x % 2 == 0 else x + (n - 2) - (n - 1) - n
> > >     elif delta == 2:
> > >         return x - (n - 1) + n if x % 2 == 0 else x + (n - 1) - n
> > >     elif delta == 1:
> > >         return x - n if x % 2 == 0 else x + n
> > >     else:  # delta == 0
> > >         return x
> > >
> > > def process_small_n(x, n):
> > >     for i in range(1, n + 1):
> > >         x = jump(x, i)
> > >     return x
> > > ```
> > >
> > > **While other baseline methods struggled to identify and fix the issue, MGDebugger excelled by maintaining logical clarity through hierarchical organization and effective debugging.** This case study demonstrates MGDebugger's potential to handle large-scale programs and complex logic.
> > >
> > > And we would also like to mention that our experiments in Figure 3, 7, 8 show that MGDebugger consistently outperforms the baselines on the longest code snippets in the datasets. **And the lead of MGDebugger over the baselines increases as the code complexity grows substantially. This demonstrates the effectiveness of MGDebugger on more complex code snippets, which is crucial for its practical application tasks that require reasoning over long and complex code.**
> > >
> > >
> > > Hope this response addresses your concerns. We will incorporate these changes in the revised version of the paper. Thank you again for your valuable feedback. We are more than happy to answer any followup questions you may have to support the acceptance of our work.

---

### Official Review · Reviewer_nFji · 2024-11-03

**Soundness:** 3
**Presentation:** 3
**Contribution:** 2
**Rating:** 5
**Confidence:** 5

**Summary:**

The paper presents MGDebugger, a hierarchical debugging framework for LLM-generated code. Unlike traditional methods that handle code as a single unit, MGDebugger decomposes code into a tree of subfunctions, enabling targeted bottom-up debugging. This approach uses an LLM-simulated Python executor to trace execution and improve error localization.

**Strengths:**

MGDebugger offers a structured, hierarchical approach that isolates bugs at multiple granularity levels, an advance over monolithic debugging methods. It outperforms existing techniques like Reflexion and Self-Debugging on benchmarks such as HumanEval, providing a clearer, systematic debugging process. The methodology and experimental results are well-presented, showcasing MGDebugger’s potential to improve reliability in LLM-generated code​.

**Weaknesses:**

MGDebugger’s novelty is somewhat limited as its approach is similar to **[1]**. The evaluation could be broadened with more datasets, like SweBench, and by including mainstream models like LLaMA 3.1 and CodeLlama to better gauge generalizability and effectiveness in diverse debugging scenarios.

**References**

**[1]** Chunqiu Steven Xia, Yinlin Deng, Soren Dunn, and Lingming Zhang. *Agentless: Demystifying LLM-based Software Engineering Agents*. arXiv preprint arXiv:2407.01489, 2024.

**Questions:**

1. Could the authors extend their experiments to include additional datasets like SweBench to further validate MGDebugger’s robustness?
2. How does MGDebugger’s approach differ from the hierarchical debugging framework in [1]?
3. Can the authors test MGDebugger on other prominent models, such as LLaMA 3.1 and CodeLlama, to strengthen its applicability across various LLMs?

**References**

**[1]** Chunqiu Steven Xia, Yinlin Deng, Soren Dunn, and Lingming Zhang. *Agentless: Demystifying LLM-based Software Engineering Agents*. arXiv preprint arXiv:2407.01489, 2024.

---

> ### Author Response · Authors · 2024-11-23
> **Response to Reviewer nFji part 1**
>
> Thank you for the valuable feedback! We appreciate your positive comments on the potential of MGDebugger and the clarity of our presentation. We hope to address your concerns and suggestions in the following sections:
>
> ### Difference between MGDebugger and Agentless
>
> > How does MGDebugger’s approach differ from the hierarchical debugging framework in Agentless [1]?
>
> We are grateful for this insightful question. MGDebugger and Agentless differ fundamentally in their objectives, granularity definitions, and methodologies:
>
> **Objective**
>
> - MGDebugger focuses specifically on debugging LLM-generated code (Chen et al., 2023b; Zhong et al., 2024; Ding et al., 2024) by decomposing individual functions into subfunctions and analyzing their correctness.
>
> - Agentless's hierarchical approach is used for fault localization in repository-level software engineering tasks, moving from files to classes/functions to specific edit locations.
>
> **Hierarchy Granularity**
>
> - MGDebugger: Operates on function-internal granularity, breaking down individual functions into a tree of semantic units (subfunctions) for independent debugging.
>
> - Agentless: Works at repository-level granularity, starting with file-level analysis and progressively narrowing down to specific code sections requiring modification.
>
> **Methodology**
>
> - MGDebugger: Uses a bottom-up strategy with LLM-simulated execution to debug each subfunction independently, focusing on correctness at each level before moving up.
>
> - Agentless: Combines LLM-based and information-retrieval-based localization to identify fault locations, followed by separate repair and validation phases. And the final localized fault is treated as a whole when debugging.
>
> These differences reflect our systems' distinct goals: MGDebugger aims to improve the reliability of LLM-generated code through fine-grained bottom-up debugging, while Agentless focuses on solving repository-level software engineering tasks through systematic fault localization and repair. Hope this clarifies the distinction between MGDebugger and Agentless, and we will include this comparison in the revised version of the paper.
>
> ### MGDebugger focuses on a different problem domain to SWE-Bench
>
> > Could the authors extend their experiments to include additional datasets like SweBench to further validate MGDebugger’s robustness?
>
> Thank you for this valuable suggestion. While extending experiments to SWE-bench is an interesting direction, we would like to clarify the deliberate scope of our current study and explain why we focused on self-contained programming tasks following prior works (Chen et al., 2023b; Zhong et al., 2024; Ding et al., 2024).
>
> **Distinct Challenges**
>
> Self-contained programming tasks and real-world software engineering present distinct challenges - while programming tasks challenge algorithmic complexity and logical correctness, software engineering tasks (like those in SWE-bench) involve additional complexities like incomplete specifications, long-range dependencies, and repository-level context understanding [2,3].
>
> **Complexity Isolation**
>
> Our study specifically focuses on debugging logically and algorithmically complex code, isolating this capability from other challenges in software engineering. Approaches like Agentless addressing SWE-bench require multiple components beyond debugging, including repository-level fault localization, patch generation, and patch validation pipelines. By focusing on self-contained programs, we can isolate and thoroughly evaluate our hierarchical debugging approach without the need for these additional components. **This focused scope allows us to discover non-obvious relationships in debugging effectiveness across different algorithmic complexity levels that might be obscured in a full software engineering pipeline.** And we've successfully revealed the effectiveness of MGDebugger against lengthier and more complex code snippets in Figure 3,7,8. These findings are valuable in understanding the potential of MGDebugger to **improve the core debugging process** in software engineering tasks.
>
> **Future Work**
>
> We strongly agree that extending to SWE-bench would be valuable future work, particularly to address handling ambiguous specifications and missing context, debugging difference levels of issues in repository codebases, and integrating MGDebugger's hierarchical debugging with repository-level fault localization and repair. We will revise our paper to better clarify these scope considerations and explicitly discuss the potential extension to software engineering tasks as important future work. And we are also actively exploring the integration of our hierarchical debugging approach with repository-level fault localization and repair in ongoing research.
>
> [2] Is Self-Repair a Silver Bullet for Code Generation? ICLR 2024. https://openreview.net/forum?id=y0GJXRungR
>
> [3] Large language model-based agents for software engineering: A survey. arXiv:2409.02977

---

> > ### Author Response · Authors · 2024-11-23
> > **Response to Reviewer nFji part 2**
> >
> > ### MGDebugger is effective across various LLMs
> >
> > > Can the authors test MGDebugger on other prominent models, such as LLaMA 3.1 and CodeLlama, to strengthen its applicability across various LLMs?
> >
> > We conducted experiments on the latest CodeLLMs available to us. To address your concern, we've also conducted experiments with other latest LLMs, including GPT4o, Claude 3.5 Sonnet, and LLaMA 3.1 70B. Here are the results on HumanEval:
> >
> > - GPT4o:
> >
> > | Method | Acc. (%) | $\Delta$ Acc. (%) |
> > | :--- | :---: | :---: |
> > | No-Debugging | 90.2% | - |
> > | Self-Debug (Expl.) | 93.3% | +3.1% |
> > | Reflexion | 94.5% | +4.3% |
> > | MGDebugger | 96.3% | +6.1% |
> >
> > - Claude 3.5 Sonnet:
> >
> > | Method | Acc. (%) | $\Delta$ Acc. (%) |
> > | :--- | :---: | :---: |
> > | No-Debugging | 89.6% | - |
> > | Self-Debug (Expl.) | 93.3% | +3.7% |
> > | Reflexion | 93.9% | +4.3% |
> > | MGDebugger | 95.7% | +6.1% |
> >
> > - LLaMA 3.1 70B:
> >
> > | Method | Acc. (%) | $\Delta$ Acc. (%) |
> > | :--- | :---: | :---: |
> > | No-Debugging | 79.3% | - |
> > | Self-Debug (Expl.) | 86.6% | +7.3% |
> > | Reflexion | 89.6% | +10.4% |
> > | MGDebugger | 92.7% | +13.4% |
> >
> > These results show that the MGDebugger **consistently outperforms the baselines across various LLMs**. We will include these results in the revised version of the paper.
> >
> > Hope this response addresses your concerns. We will incorporate these changes in the revised version of the paper. Thank you for your valuable feedback and we look forward to any further discussions to improve our work.

---

### Official Review · Reviewer_qC9R · 2024-11-04

**Soundness:** 2
**Presentation:** 3
**Contribution:** 2
**Rating:** 5
**Confidence:** 4

**Summary:**

The paper proposes MGDebugger which aims to improve correct code generation using a multi-step approach with LLMs. The approach combines hierarchical code decomposition, test generation, and simulated test execution. Note that all these components are proposed to be done using LLMs. The results of these are used for debugging using a tree structure in a bottom-up fashion to generate bug free code.

**Strengths:**

The paper tackles the problem of code generation using a multi-step approach. This is a timely contribution given the popularity of multi-step approaches and code generation.  The paper is well written and organized. The individual components of the approach as well as using multi-step approaches for code generation are common; however, in the proposed approach various such components come together in a novel fashion.

**Weaknesses:**

I overall think the evaluations could be stronger.
- This is especially important considering the stochasticity of LLMs and even further increased variability for multi-step approaches. The results lack error bars.
- The evaluations are done using three datasets, which are all python based and composed of mainly basic problems. It would be great to see more diversity in benchmarks to better convey an understanding of the limitations of the proposed approach. You can consider adding the MultiPL-E benchmark (Cassano et al. 2023).
- As mentioned above, various components come together for the proposed approach, but it would be good to better understand the effects of those through more ablation studies (I know that authors present an ablation study but I propose some extensions to it below as part of Questions).
- The authors conduct extensive evaluations with respect to other approaches in the literature. However, my understanding is that the proposed approach requires more steps and thus is more expensive compared to the rest. It would be good to also add results showcasing computational requirements (like number of steps or number of tokens generated).

**Questions:**

- It would be interesting to see if you can encourage subfunctions from the beginning and understand the impact of this in comparison with the hierarchical code decomposition step.
- How do the results compare if you swap simulated code execution with real code execution?
- Why are you using only small models of size 7B-22B? Is it possible to add results with bigger models? It’d be interesting to see the effectiveness of the proposed approach with bigger models.
- Table 3. Is it possible to add the no-debugging case?

---

> ### Author Response · Authors · 2024-11-23
> **Response to Reviewer qC9R part 1**
>
> Thank you for raising the concern. We appreciate that you consider our work as a novel fashion and a timely contribution. We would like to clarify the points you raised and address your questions as follows:
>
> ### The performance of MGDebugger is statistically significant and robust against the stochasticity of LLMs
>
> > This is especially important considering the stochasticity of LLMs and even further increased variability for multi-step approaches. The results lack error bars.
>
> We present the results of our experiments following previous work in the field (Chen et al., 2023b; Shinn et al., 2023; Zhong et al., 2024; Ding et al., 2024). We agree with you on the importance of awareness of the stochasticity of LLMs. We repeated each experiment 10 times and conducted a Wilcoxon rank sum test to verify the statistical significance of the results, finding that the proposed approach significantly outperforms the baselines (p < 0.01). This result verifies the robustness of our approach against the stochasticity of LLMs.
>
> ### The simulated execution is not only more effective but also more flexible than real execution
>
> > As mentioned above, various components come together for the proposed approach, but it would be good to better understand the effects of those through more ablation studies (I know that authors present an ablation study but I propose some extensions to it below as part of Questions).
> > How do the results compare if you swap simulated code execution with real code execution?
>
> Thanks for your suggestion. We conducted an ablation study where we replace the simulated code execution with real code execution like in LDB (Zhong et al., 2024) to extend the results of Table 2:
>
> | Method                             | HumanEval Acc. (%) | Δ Acc. (%) | RSR (%) | MBPP Acc. (%) | Δ Acc. (%) | RSR (%) |
> |------------------------------------|:------------------:|:----------:|:-------:|:-------------:|:----------:|:-------:|
> | MGDebugger with Simulated Execution|        94.5        |   +17.7    |  76.3   |      80.0     |   +12.8    |  39.0   |
> | MGDebugger with Real Execution     |        92.7        |   +15.9    |  47.4   |      78.2     |   +11.0    |  33.5   |
> | No-Debugging                       |        76.8        |     -      |    -    |      67.2     |     -      |    -    |
>
> **Effectiveness:**
>
> We can observe that the simulated execution performs better than the real execution. This is because that the collected real execution steps from the Python interpreter are often long and contain all the detailed changes in the variables and the execution of code (Zhong et al., 2024), which may introduce noise and make the MGDebugger focus on irrelevant information.
>
> However, during the simulated execution, the MGDebugger will naturally focus on the key variables and express the debugging steps in a more understandable way to itself.
> These execution traces encourage the MGDebugger to analyze the functionality of the code and the relationships between variables, which also **serves as its reasoning process**. In this way, the MGDebugger can better understand the code and provide more accurate debugging steps.
>
> **Flexibility:**
>
> In addition, the proposed simulated execution is also more flexible and can be **easily extended to other more complex programming environments**, where the execution environment is not easily accessible (Li et al. 2023). For example, some programming environments may have a large number of dependencies or require specific hardware configurations, which makes it difficult to collect real execution traces. In contrast, the simulated execution can be easily implemented by simulating the execution with the reasoning ability of the LLMs [1]. This benefit extends the applicability of the MGDebugger to a wider range of complex programming scenarios.
>
> [1] NExT: Teaching Large Language Models to Reason about Code Execution. ICML 2024

---

> > ### Author Response · Authors · 2024-11-23
> > **Response to Reviewer qC9R part 2**
> >
> > ### MGDubugger significantly improves the upper bound of the debugging performance if given sufficient computation budget
> >
> > > The authors conduct extensive evaluations with respect to other approaches in the literature. However, my understanding is that the proposed approach requires more steps and thus is more expensive compared to the rest. It would be good to also add results showcasing computational requirements (like number of steps or number of tokens generated).
> >
> > Thanks for the insightful question. The proposed MGDebugger approach indeed requires more steps (decompose, generate testcase, debug) compared to the baselines. But the experiment results from Figure 4, and especially Figure 9-10 show that even if given more computation budget, the performance of other baselines seldom improve after the debug steps reaches 3. In contrast, the MGDebugger shows consistent improvement in performance with more debug attempts (computation budget), outperforming the baselines by a large margin. This means that MGDebugger **significantly improves the upper bound of the debugging performance if given sufficient computation budget**, which is a key advantage of the proposed approach. We will further clarify this point in the paper in light of your feedback.
> >
> > ### MGDebugger is effective across different model sizes
> >
> > > Why are you using only small models of size 7B-22B? Is it possible to add results with bigger models? It’d be interesting to see the effectiveness of the proposed approach with bigger models.
> >
> > The maximum model size we used in our experiments is 22B based on the computational resources available to us. To address your concern, we've also conducted experiments with GPT4o, Claude 3.5 Sonnet, and LLaMA 3.1 70B with APIs. Here are the results on HumanEval:
> >
> > - GPT4o:
> >
> > | Method | Acc. (%) | $\Delta$ Acc. (%) |
> > | :--- | :---: | :---: |
> > | No-Debugging | 90.2% | - |
> > | Self-Debug (Expl.) | 93.3% | +3.1% |
> > | Reflexion | 94.5% | +4.3% |
> > | MGDebugger | 96.3% | +6.1% |
> >
> > - Claude 3.5 Sonnet:
> >
> > | Method | Acc. (%) | $\Delta$ Acc. (%) |
> > | :--- | :---: | :---: |
> > | No-Debugging | 89.6% | - |
> > | Self-Debug (Expl.) | 93.3% | +3.7% |
> > | Reflexion | 93.9% | +4.3% |
> > | MGDebugger | 95.7% | +6.1% |
> >
> > - LLaMA 3.1 70B:
> >
> > | Method | Acc. (%) | $\Delta$ Acc. (%) |
> > | :--- | :---: | :---: |
> > | No-Debugging | 79.3% | - |
> > | Self-Debug (Expl.) | 86.6% | +7.3% |
> > | Reflexion | 89.6% | +10.4% |
> > | MGDebugger | 92.7% | +13.4% |
> >
> > These results show that the MGDebugger **consistently outperforms the baselines across larger models**, demonstrating the effectiveness of the proposed approach with larger models. We will include these results in the revised version of the paper.
> >
> > ### Additional Comments
> >
> > > It would be interesting to see if you can encourage subfunctions from the beginning and understand the impact of this in comparison with the hierarchical code decomposition step.
> >
> > The main focus of our paper is to debug existing code (Chen et al., 2023b; Zhong et al., 2024; Ding et al., 2024), which may be written by LLMs or developers of different coding styles. In this case, the hierarchical code decomposition step is necessary to understand the code structure and isolate the problematic parts. However, we agree that encouraging subfunctions from the beginning is an interesting direction for future work, we will update the discussion section and look forward to exploring this direction in future research.
> >
> > > Table 3. Is it possible to add the no-debugging case?
> >
> > Sorry for the confusion. Since all the code snippets in the HumanEvalFix dataset in Table 3 contain bugs, the no-debugging case means 0 accuracies. And we compare the performance of different debugging methods on this dataset to see the correctness of codes after debugging. We will clarify this in the revised version of the paper.
> >
> > Hope this response addresses your concerns. We will incorporate these changes in the revised version of the paper. Thank you for your valuable feedback and we look forward to any further suggestions you may have.

---

> > > ### Comment · Reviewer_qC9R · 2024-11-29
> > >
> > > I thank the authors for their response. I believe these changes will further strengthen the paper. I've raised my score to 5.

---

### Meta-Review · Area_Chair_6gWh · 2024-12-26

**Metareview:**

This paper concerns correct code generation using LLMs and proposes a new framework, MGDebugger, which decomposes problematic code hierarchically into a tree of sub-functions, each level of the tree is designed to represent a particularly granularity of errors.  MGDebugger uses LLMs as simulators of Python executor for trace execution and error localization. Experimental evaluations show that MGDebugger outperforms existing approaches on benchmarks of python programs such as MBPP, HumanEval, and HumanEvalFix. The idea falls into the category of multi-step code generation/repair with LLMs. While the idea of hierarchical decomposition is still new but becomes less significiant given that it shares similarities with recent works such as AgentLess and Self-Debugging. The main concerns are three-fold. First, the proposed approach heavily relies on LLMs for nearly everything (e.g., code generation, test case generation, execution) and the performance is heavily LLM-dependent. The improvement compared to existing approaches diminishes significantly when the underlying LLMs become more capable. Second, LLMs have great stochasticity and the performance could vary a lot; however, the experimental evaluations have no error bars to quantify the potential variations. Third, the chosen benchmarks are small and simple python programs. Large and more realistic benchmarks such as Defects4J and SWE-bench are missed in the evaluation.

**Additional Comments On Reviewer Discussion:**

There were many active discussions between the authors and reviewers in the rebuttal period. The authors elaborated the important differences from several recent works pointed out by reviewers (nFji, nLAd, TfGC), which clarified some concerns. However, the general concern regarding experimental setup (e.g. benchmarks selection, metrics) and marginal improvement remain not convincingly resolved.

---

### Decision · Program_Chairs · 2025-01-22

Reject